# EMR-MERGING: Tuning-Free High-Performance Model Merging

**Chenyu Huang**[1†]**, Peng Ye**[1,3†‡]**, Tao Chen**[1*]**, Tong He**[2]**, Xiangyu Yue**[3]**, Wanli Ouyang**[3]
[1] Fudan University     [2] Shanghai AI Laboratory
[3] The Chinese University of Hong Kong
`cyhuang24@m.fudan.edu.cn`

## Abstract

The success of pretrain-finetune paradigm brings about the release of numerous model weights. In this case, merging models finetuned on different tasks to enable a single model with multi-task capabilities is gaining increasing attention for its practicability. Existing model merging methods usually suffer from (1) significant performance degradation or (2) requiring tuning by additional data or training. In this paper, we rethink and analyze the existing model merging paradigm. We discover that using a single model's weights can hardly simulate all the models' performance. To tackle this issue, we propose ELECT, MASK & RESCALE-MERGING (EMR-MERGING). We first (a) elect a unified model from all the model weights and then (b) generate extremely light-weight task-specific modulators, including masks and rescalers, to align the direction and magnitude between the unified model and each specific model, respectively. EMR-MERGING is tuning-free, thus requiring no data availability or any additional training while showing impressive performance. We find that EMR-MERGING shows outstanding performance compared to existing merging methods under different classical and newly-established settings, including merging different numbers of vision models (up to **30**), NLP models, PEFT models, and multi-modal models. [1]

## 1 Introduction

With the rapid development of deep learning, different model architectures [36, 22, 71, 88] are proposed, along with multiple training strategies [89, 86]. Pre-trained models' capabilities are enhanced, thus showing increasing significance [54, 22, 7, 19]. Finetuning models on downstream tasks from a pre-trained model has become a standard paradigm in both NLP and vision fields [20, 51, 19, 22, 5, 87], which usually leads to improved performance with less labeled data. With the development of open-source repositories such as Huggingface [79], timm [77], and torchvision [44], the number of pre-trained and finetuned checkpoints exponentially rise. However, applying individual models to different tasks results in high storage and deployment costs. Multi-task learning (MTL) partially solves this problem by jointly training a model using multiple datasets [70, 93, 95], but it suffers from (i) high computational costs and (ii) data unavailability due to privacy [33]. Recently, model merging attempts to solve these drawbacks by combining weights instead of additional training, thus showing vital significance and broad application prospects.

A simple strategy of model merging is averaging the model weights [80], but it usually causes obvious performance degradation, as shown in Fig. 1. To this end, there are multiple model merging methods proposed to improve the performance of the merged model, which can be roughly divided into three

---

[*]Corresponding Author(eetchen@fudan.edu.cn).    [†]Equal Contribution.    [‡]Project Lead.

[1]Our code is available at https://github.com/harveyhuang18/EMR_Merging.

38th Conference on Neural Information Processing Systems (NeurIPS 2024).

categories: (i) *Weighted averaging of model weights* include Fisher-Merging [46] and RegMean [33]. They use pre-computed Fisher information matrices [23] and inner-product matrices [33] to tune the coefficients for weighted averaging. (ii) *Task vector-based methods* that add task vectors together instead of model weights, include Task Arithmetic [30], Ties-Merging [84], and AdaMerging [85]. Ties-Merging handles the interference issue and AdaMerging adaptively tunes the merging coefficients. (iii) *Pre-processing techniques* include DARE [90]. It reduces interference by dropping most elements and rescaling the others in task vectors. Despite the promising results, there are two unresolved problems with the existing model merging methods: (1) The performance gap between the merged model and individual models or MTL is still obvious, as shown in Fig. 1. (2) The performance improvement of existing methods depends on tuning by data or training, as shown in Tab. 1.

To boost the performance of model merging, we rethink and analyze the existing model merging paradigm. We discover that the goal of all the existing methods is to obtain a single model applicable to all the $N$ tasks, as follows:

$$W_M = \mathcal{M}\left([W_1..W_N]\right), \quad (1)$$

where $[W_1..W_N]$ are the model weights to be merged, $\mathcal{M}$ denotes the merging function, and $W_M$ is the merged model weight. This paradigm may inevitably lead to a non-negligible gap between the merged model and each individual model, especially when there are numerous models or models on challenging tasks. We argue that using a single model weight to simulate all the model weights is sub-optimal. To tackle this issue, we propose a brand new merging paradigm: We first extract a unified model weight from all the models' weights, and then we calculate and store significant but lightweight task-specific parts of each model weight. This process can be written as:

$$W_{uni}, [E_1..E_N] = \mathcal{M}'\left([W_1..W_N]\right), \quad (2)$$

where $W_{uni}$ represents the common and shared part of all model weights and $[E_1..E_N]$ denote the task-specific parts of each model weight. $\mathcal{M}'$ is the revised merging function following our paradigm.

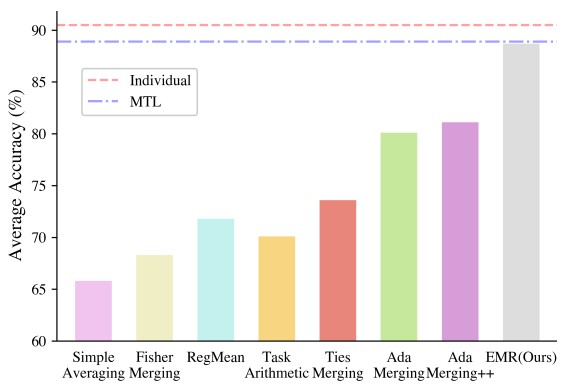

Figure 1: The average accuracy of the multi-task performance of different model merging methods on eight vision tasks. Among all the merging methods, our EMR-MERGING is the only one comparable to the performance of MTL and even individual models.

Table 1: Prerequisites for each method's working.

| Methods | Training-Data Tuning | Valid-Data Tuning inputs | Valid-Data Tuning labels | Tuning by Training |
|---|---|---|---|---|
| Weight Averaging | × | × | × | × |
| Traditional MTL | ✓ | × | × | ✓ |
| Fisher-Merging [46] | × | ✓ | × | × |
| RegMean [33] | × | ✓ | × | × |
| Task Arithmetic [30] | × | ✓ | ✓ | × |
| Ties-Merging [84] | × | ✓ | ✓ | × |
| AdaMerging [85] | × | ✓ | × | ✓ |
| **EMR-Merging**(Ours) | × | × | × | × |

Based on the above paradigm, we propose EMR-MERGING (ELECT, MASK & RESCALE-MERGING). We first elect a unified model from all the model weights. The election strategy is choosing the maximum absolute value of each parameter on the specified sign direction to minimize interference and avoid additional tuning. Then we generate additional lightweight task-specific modulators, including masks and rescalers. Their functions are respectively to align the direction and magnitude of the unified model with the original task-specific model. We find that applying the task-specific modulators to the unified model can better approximate the task-specific model, thus improving performance. The detailed process, theoretical and empirical analysis of the proposed method are illustrated in Section 3. By applying our method, the performance of model merging is significantly enhanced and is comparable to MTL or individual models, as shown in Fig. 1. Meanwhile, EMR-MERGING requires no data, tuning, or any additional training, as shown in Tab. 1.

We first demonstrate the effectiveness of the proposed EMR-MERGING under the existing setting of (1) merging Vision Transformer (ViT) [22] models of different sizes on 8 vision tasks, (2) merging parameter-efficient finetuning (PEFT) models on 11 language tasks, and (3) merging GPT-2 [55] models on 7 language tasks. Our method shows significant performance improvement under these settings, even when compared to the strongest baseline. We further validate the method's effectiveness under newly-established and more challenging settings including: (4) merging ViTs on **30** vision

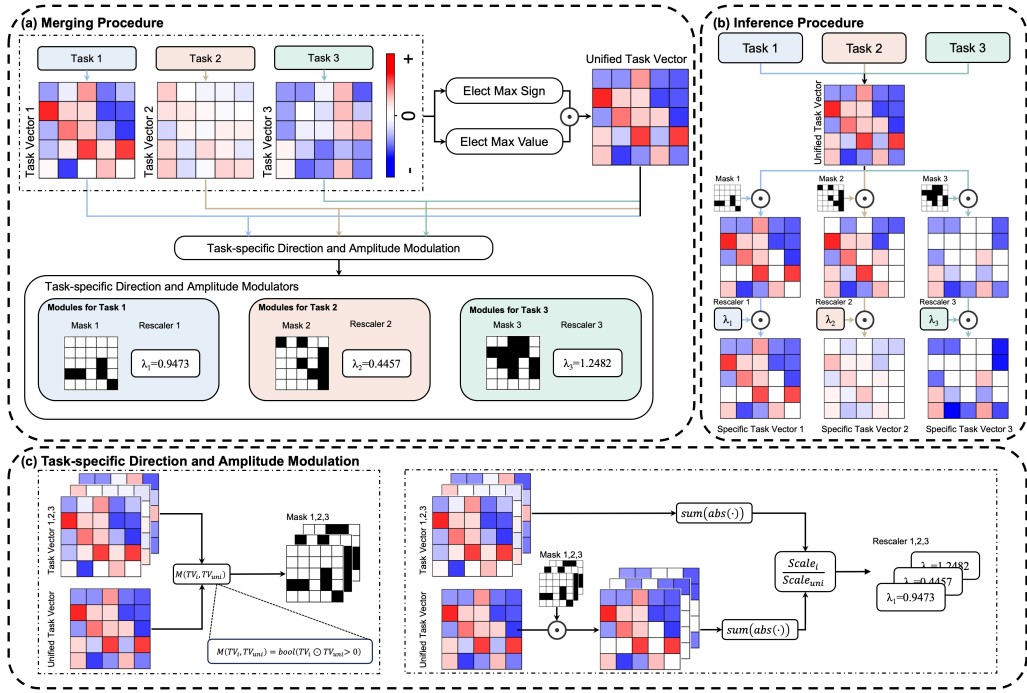

Figure 2: **Framework overview**. In the (a) Merging Procedure, we merge task-specific vectors into a unified task vector and lightweight task-specific modulators to modulate direction and amplitude. During the (b) Inference Procedure, we apply the corresponding mask and rescaler to the unified task vector to obtain a specific task vector. The process of (c)Task-specific Direction and Amplitude Modulation includes obtaining task-specific masks and scalers.

tasks, (5) merging RoBERTa [43] models on 8 NLP tasks, and (6) merging BEiT3 [75] models on 5 multi-modal tasks.

Our contributions can be summarized as: (1) We propose a novel merging method called EMR-MERGING, which merges task-specific models into a unified model and lightweight task-specific modulators (i.e., masks and rescalers), requiring no data, tuning, or additional training. (2) The proposed EMR-MERGING is simple-but-effective, and its effectiveness is validated on various classical benchmarks and newly-established benchmarks under various vision, NLP, PEFT, and multi-modal settings. (3) We show that the masks and rescalers of EMR-MERGING for aligning task-specific direction and amplitude of task vectors are applicable to most kinds of merging methods.

## 2 Related Work

**Model Merging** obtains a model using the existing task-specific model weights instead of training [33, 30, 84, 85, 66, 90, 46]. Simply averaging [80] usually causes severe performance degradation. Various methods are proposed to handle this problem. Fisher-Merging [46] and RegMean [33] use fisher information matrices [23] and inner-product matrices [33] to calculate the merging coefficients for weighted merging. However, they require additional matrices released by model owners or manually computed. Task Arithmetic [30] merges models by adding together task vectors, which is the difference between the finetuned and pre-trained models. Ties-Merging [84] and AdaMerging [85] are based on task vectors. Ties-Merging resolves interference and AdaMerging adaptively learns the merging coefficients. However, the performance of Task Arithmetic and Ties-Merging highly depends on manually tuning the merging coefficients and AdaMerging needs additional training to obtain them. DARE [90] reduces interference by randomly dropping most elements and rescaling the remaining ones in each task vector before merging. However, DARE's performance is only validated under the setting of merging a limited number of tasks and the performance gain is also limited. In addition, all the existing methods merge models into a single one, and have not been verified under experimental settings of more models to merge, models on more difficult tasks, and

multi-modal models. In this paper, we propose EMR-MERGING, which requires no tuning while showing impressive performance under various settings.

**Multi-Task Learning** trains a single model using training data from multiple tasks together [70, 93, 95]. MTL typically necessitates access to the labeled data of multiple tasks for training the model from scratch. Though enabling the model multi-task capabilities, MTL suffers from not only (i) the expensive computational cost for training, especially for large models, but also (ii) the limited data availability due to data privacy [85]. In comparison, model merging solves the mentioned problems by combining the model weights without using training data or additional training, thus obtaining a multi-task model while sharply reducing the costs.

**Supervised Finetuning** from pre-trained models on down-stream tasks is becoming a standard paradigm in both NLP and vision fields [20, 51, 19, 22, 5]. Depending on whether all the parameters of models are adjusted, SFT can be divided into conventional full finetuning (FFT) and parameter-efficient finetuning (PEFT), which is proposed to reduce the number of trainable parameters for downstream tasks by adjusting the inserted small modules called adapters while keeping the whole model frozen [28, 29, 42]. PEFT is becoming the prevailing method to adapt pre-trained large models because of its efficiency [94]. There are a large number of pre-trained, full finetuned model weights, and PEFT module weights available on public repositories [79, 77, 44]. In this paper, the proposed EMR-MERGING is based on the common pretrain-finetune paradigm and we show the applicability of our method to both full finetuned models and PEFT modules.

# 3 Method

## 3.1 Motivation

Given $N$ tasks $[T_1..T_N]$, the goal of model merging is to obtain a model applicable to all the tasks using finetuned models $[W_1..W_N]$ from the same pre-trained model $W_{pre}$ on each task. Existing methods focus on merging the models into a single model $W_M$. Please check Appendix C for detailed information on the existing merging methods. However, a single model can hardly represent all the model weights, thus causing severe performance drops. We discover that the combination of a unified task vector and lightweight task-specific modulators can settle this issue to a significant extent by approximating the task-specific vectors better without any additional tuning. The size of proposed task-specific modulators is discussed in Section 4.4, which is much smaller than that of a model.

## 3.2 ELECT, MASK & RESCALE-MERGING

The overall framework of EMR-MERGING is shown in Fig. 2. We follow the setting of task vector-based methods [30, 84, 85] and we merge models using task vectors. For task $T_i$, $i \in [1..N]$, the corresponding task vector is defined as $\tau_i = W_i - W_{pre}$, where $\tau_i \in \mathbb{R}^d$.

**Electing a unified task vector** We first create an aggregate elected sign vector $\gamma_{uni} = sgn(\sum_{t=1}^N \tau_t)$ by choosing the sign with the higher total magnitude of each parameter across all relevant task vectors. Then we choose the maximum absolute value of each parameter with the sign consistent with $\gamma_{uni}$ from all the task vectors and obtain absolute value vector $\epsilon_{uni} \in \mathbb{R}^d$. By combining $\gamma_{uni}$ and $\epsilon_{uni}$, the unified task vector can be obtained by $\tau_{uni} = \gamma_{uni} \odot \epsilon_{uni}$. The electing procedure can reserve the maximum amplitude and sign information shared by the task vectors, thereby maximally reducing interference. The unified task vector $\tau_{uni}$ corresponds the $W_{uni}$ in Eq. 2. Before being applied to task $T_i$, the $\tau_{uni}$ needs to be modulated in advance by task-specific modulators, which are corresponding to $E_i$ in Eq. 2. The generation of task-specific modulators is described below:

**Task-specific masks.** Next, we compare the unified task vector $\tau_{uni}$ with each task vector $\tau_i$. The task-specific mask $M_i = (\tau_i \odot \tau_{uni} > 0)$ for task $i$ sets the elements whose signs are not correspondent with $\tau_{uni}$ to zero and the rest to one. The function of the masks is to align the direction of the unified model with the task-specific model. The masks share the same structure with the task-specific models but due to their 1-bit nature, the size of a mask is much smaller than that of a task vector.

**Task-specific Rescalers** Then, for each task, we compute a rescaler parameter to keep the average absolute value of the elements in $\tau_t$ and $M_t \odot \tau_{uni}$ equal. The function of the rescalers $\lambda_i = \frac{sum(abs(\tau_i))}{sum(abs(M_i \odot \tau_F))}$ is to align the parameter magnitude of the unified model with the task-specific

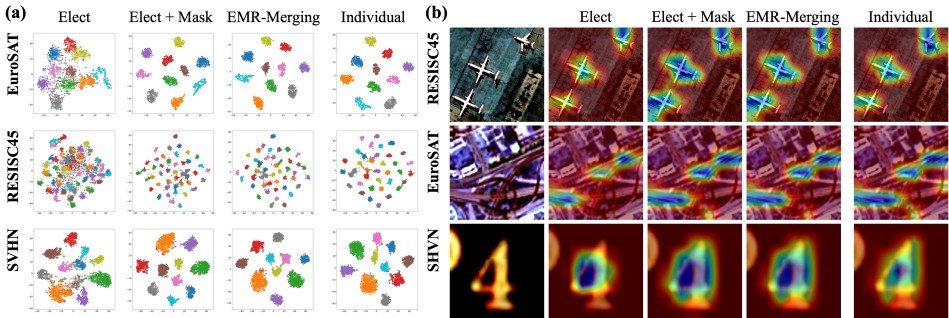

Figure 3: Partial (a) t-SNE and (b) Grad-CAM visualization results of EMR-MERGING's procedures.

model. The significance of rescaling is also reported by DARE [90], which claims that after dropping most elements in a task vector, rescaling the rest leads to better results compared to not.

Before being applied to a task, a task-specific modulation is required to be conducted to the unified task vector. After that, we add it to the pre-trained parameter values $W_{pre}$. The inference steps of applying the merged model to task $t$ are as follows: $\hat{W}_t = W_{pre} + \hat{\tau}_t$, where $\hat{\tau}_t = \lambda_t \cdot M_t \odot \tau_{uni}$. It should be noted that during the whole process, no additional tuning is needed, thus requiring no data or additional training. We summarize the algorithm flow in Appendix A.

### 3.3 Theoretical analysis

Our goal is to merge model weights by minimizing the distance between the merged model $W_{uni}$ and each individual model $W_i$, where the distance can be calculated by:

$$Dis = \frac{\sum_{i=1}^{N} \|W_i - W_{uni}\|^2}{N} = \frac{\sum_{i=1}^{N} \|\tau_i - \tau_{uni}\|^2}{N} \tag{3}$$

where $\tau_i$ refers to the task vector for task $T_i$ and $\tau_{uni}$ is the unified task vector.

**Analysis 1: Effectiveness of Masks.** After applying the masks $M_i = (\tau_i \odot \tau_{uni} > 0)$ to the unified model $\tau_{uni}$, the distance $Dis^M$ can be formulated as:

$$Dis^M = \frac{\sum_{i=1}^{N} \|\tau_i - M_i \odot \tau_{uni}\|^2}{N} \leq Dis \tag{4}$$

where $Dis$ refers to the distance before applying the masks. Eq. 4 demonstrates that the distance between the merged model and each individual model can be reduced after applying the masks.

**Analysis 2: Effectiveness of Rescalers.** After applying the rescalers $\lambda_i = \frac{sum(abs(\tau_i))}{sum(abs(M_i \odot \tau_F))}$ to the masked task vectors $M_i \cdot \tau_{uni}$, the distance $Dis^{M,\lambda}$ is formulated as:

$$Dis^{M,\lambda} = \frac{\sum_{i=1}^{N} \|\tau_i - \lambda_i \cdot M_i \odot \tau_{uni}\|^2}{N} \leq Dis^M \tag{5}$$

Eq. 5 demonstrates that the distance between the merged model and each individual model can be minimized after applying the rescalers. Please check Appendix B for detailed proof.

### 3.4 Empirical analysis

In Fig. 3, we visualize partial results of merging eight ViT-B/32 models on different tasks using t-SNE [69] and Grad-CAM [61]. It can be seen that each procedure of EMR-MERGING can help improve the performance of the merged model and perform closer to individual models. Specifically, a more obvious distinction is shown in t-SNE and a more precise target is focused by Grad-CAM. Please check Section 4.1.1 for experimental details and Appendix E for more visualization results.

Table 2: Multi-task performance when merging ViT-B/32 models on eight tasks.

| Methods | SUN397 | Cars | RESISC45 | EuroSAT | SVHN | GTSRB | MNIST | DTD | Avg Acc |
|---|---|---|---|---|---|---|---|---|---|
| Individual | 75.3 | 77.7 | 96.1 | 99.7 | 97.5 | 98.7 | 99.7 | 79.4 | 90.5 |
| Traditional MTL | 73.9 | 74.4 | 93.9 | 98.2 | 95.8 | 98.9 | 99.5 | 77.9 | 88.9 |
| Weight Averaging | 65.3 | 63.4 | 71.4 | 71.7 | 64.2 | 52.8 | 87.5 | 50.1 | 65.8 |
| Fisher Merging [46] | 68.6 | 69.2 | 70.7 | 66.4 | 72.9 | 51.1 | 87.9 | 59.9 | 68.3 |
| RegMean [33] | 65.3 | 63.5 | 75.6 | 78.6 | 78.1 | 67.4 | 93.7 | 52.0 | 71.8 |
| Task Arithmetic [30] | 63.8 | 62.1 | 72.0 | 77.6 | 74.4 | 65.1 | 94.0 | 52.2 | 70.1 |
| Ties-Merging [84] | 64.8 | 62.9 | 74.3 | 78.9 | 83.1 | 71.4 | 97.6 | 56.2 | 73.6 |
| AdaMerging [85] | 64.5 | 68.1 | 79.2 | 93.8 | 87.0 | 91.9 | 97.5 | 59.1 | 80.1 |
| AdaMerging++ [85] | 66.6 | 68.3 | 82.2 | 94.2 | 89.6 | 89.0 | 98.3 | 60.6 | 81.1 |
| **EMR-MERGING (Ours)** | **75.2** | **72.8** | **93.5** | **99.5** | **96.9** | **98.1** | **99.6** | **74.4** | **88.7** |

Table 3: Multi-task performance when merging ViT-L/14 models on eight tasks.

| Methods | SUN397 | Cars | RESISC45 | EuroSAT | SVHN | GTSRB | MNIST | DTD | Avg Acc |
|---|---|---|---|---|---|---|---|---|---|
| Individual | 82.3 | 92.4 | 97.4 | 100 | 98.1 | 99.2 | 99.7 | 84.1 | 94.2 |
| Traditional MTL | 80.8 | 90.6 | 96.3 | 96.3 | 97.6 | 99.1 | 99.6 | 84.4 | 93.5 |
| Weight Averaging | 72.1 | 81.6 | 82.6 | 91.9 | 78.2 | 70.7 | 97.1 | 62.8 | 79.6 |
| Fisher Merging [46] | 69.2 | 88.6 | 87.5 | 93.5 | 80.6 | 74.8 | 93.3 | 70.0 | 82.2 |
| RegMean [33] | 73.3 | 81.8 | 86.1 | 97.0 | 88.0 | 84.2 | 98.5 | 60.8 | 83.7 |
| Task Arithmetic [30] | 74.1 | 82.1 | 86.7 | 93.8 | 87.9 | 86.8 | 98.9 | 65.6 | 84.5 |
| Ties-Merging [84] | 76.5 | 85.0 | 89.3 | 95.7 | 90.3 | 83.3 | 99.0 | 68.8 | 86.0 |
| AdaMerging [85] | 79.0 | 90.3 | 90.8 | 96.2 | 93.4 | 98.0 | 99.0 | 79.9 | 90.8 |
| AdaMerging++ [85] | 79.4 | 90.3 | 91.6 | 97.4 | 93.4 | 97.5 | 99.0 | 79.2 | 91.0 |
| **EMR-MERGING (Ours)** | **83.2** | **90.7** | **96.8** | **99.7** | **97.9** | **99.1** | **99.7** | **82.7** | **93.7** |

In Fig. 4, we compare sign conflicts, L2 distance, and cosine similarity between the merged model weights obtained by different merging methods and the task-specific model weights. It can be seen that EMR-MERGING significantly reduces sign conflicts and L2 distance and improves the cosine similarity, indicating that EMR-MERGING approximates each task-specific model weight effectively. The configuration of Fig. 4 can be found in Appendix F.

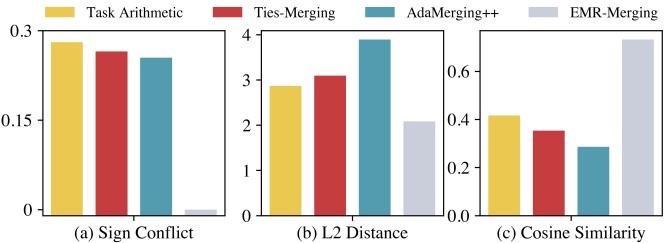

Figure 4: Comparison of (a) sign conflicts, (b) L2 distance, and (c) cosine similarity of model weights obtained by different methods and task-specific model weights.

## 4 Experiment Validation

**Baseline methods.** We compare the proposed EMR-MERGING with: (1) Individual Models, (2) Traditional MTL, (3) Weight Averaging, (4) Fisher Merging [46], (5) RegMean [33], (6) Task Arithmetic [30], (7) Ties-Merging [84], (8) AdaMerging [85]. For more details about baseline methods, please check Appendix C.

### 4.1 Merging vision models

#### 4.1.1 Merging 8 ViTs.

**Settings.** We follow the setting from Task Arithmetic [30], Ties-Merging [84], and AdaMerging [85]. We employ ViT-B/32 and ViT-L/14, two variants of CLIP [54] models' visual encoders, as the pre-trained models. The performance of each method is evaluated by eight image classification tasks, including SUN397 [83], Cars [35], RESISC45 [10], EuroSAT [27], SVHN [91], GTSRB [65], MNIST [38], and DTD [11]. All the datasets are evaluated by accuracy.

**Results.** The experimental results of merging ViT-B/32 and ViT-L/14 on eight tasks are shown in Tab. 2 and Tab. 3. We observe that EMR-MERGING shows significant performance improvement

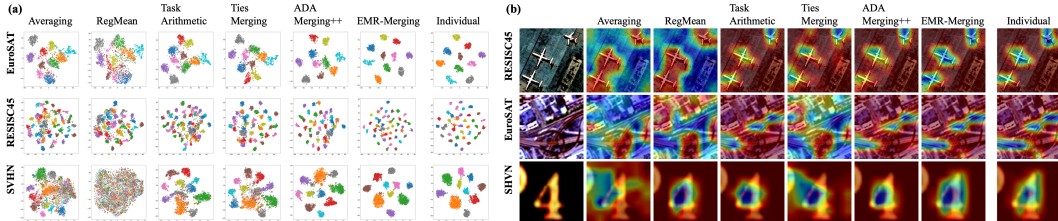

Figure 5: Partial visualization results of different merging methods, (a) t-SNE and (b) Grad-CAM.

Table 4: Task-specific and average performance when merging ViT-B/16 models on **30** tasks.

| Task-specific Acc | MNIST | Cifar-10 | Vegetables | Food-101 | Kvasir-v2 | Intel-Images | Cars | EuroSAT | Weather | Cats and Dogs |
|---|---|---|---|---|---|---|---|---|---|---|
| Individual | 99.22 | 97.88 | 100.00 | 87.93 | 94.31 | 94.63 | 85.96 | 99.04 | 98.22 | 99.05 |
| Weight Averaging | 27.63 | 42.91 | 83.20 | 68.02 | 25.27 | 82.40 | 7.74 | 24.37 | 61.06 | 91.28 |
| RegMean [33] | 90.71 | 89.65 | 99.10 | 76.14 | 71.00 | 93.60 | 16.28 | 74.13 | 86.62 | 98.54 |
| Task Arithmetic [30] | 30.81 | 59.86 | 91.97 | 73.06 | 31.05 | 89.03 | 9.34 | 31.25 | 74.56 | 93.61 |
| Ties-Merging [84] | 23.21 | 42.82 | 92.31 | 73.22 | 21.09 | 89.39 | 5.30 | 10.98 | 72.86 | 91.88 |
| AdaMerging [85] | 81.22 | 87.54 | 97.97 | 75.23 | 22.76 | 91.02 | 0.42 | 44.60 | 89.13 | 96.91 |
| **EMR-Merging (Ours)** | **98.99** | **96.69** | **99.97** | **85.05** | **93.67** | **95.27** | **72.48** | **96.24** | **97.76** | **99.27** |
| | Dogs | Fashion | Pet | LandScape | Flowers | STL-10 | CUB-200-2011 | EMNIST | DTD | RESISC45 |
| Individual | 85.16 | 93.26 | 92.23 | 86.83 | 98.19 | 99.07 | 84.79 | 94.67 | 71.76 | 98.90 |
| Weight Averaging | 47.80 | 20.46 | 31.26 | 73.14 | 68.97 | 37.74 | 37.66 | 7.73 | 14.63 | 13.56 |
| RegMean [33] | 42.89 | 83.42 | 34.62 | 83.64 | 95.26 | 78.94 | 49.78 | 48.67 | 30.53 | 34.66 |
| Task Arithmetic [30] | 47.65 | 37.11 | 33.24 | 79.59 | 80.68 | 39.66 | 41.86 | 11.05 | 14.73 | 15.50 |
| Ties-Merging [84] | 26.03 | 27.05 | 12.84 | 78.27 | 34.33 | 6.17 | 31.28 | 5.61 | 3.71 | 6.79 |
| AdaMerging [85] | 53.09 | 76.76 | 48.34 | 81.98 | 95.69 | 68.91 | 48.19 | 18.02 | 16.68 | 24.83 |
| **EMR-Merging (Ours)** | **81.89** | **92.41** | **87.15** | **86.17** | **97.66** | **98.41** | **74.91** | **92.03** | **60.05** | **93.01** |
| | MangoLeafBD | Beans | Cifar-100 | GTSRB | SVHN | SUN397 | KenyanFood13 | Animal-10N | Garbage | Fruits-360 |
| Individual | 100.00 | 97.73 | 89.85 | 95.74 | 96.22 | 78.98 | 85.53 | 92.52 | 93.36 | 99.63 |
| Weight Averaging | 68.58 | 70.98 | 77.98 | 15.00 | 10.88 | 57.42 | 33.55 | 46.00 | 22.89 | 5.38 |
| RegMean [33] | 98.10 | 92.58 | 82.59 | 56.96 | 66.13 | 58.58 | 57.11 | 68.74 | 65.31 | 19.79 |
| Task Arithmetic [30] | 87.02 | 84.62 | 80.20 | 37.01 | 17.41 | 55.88 | 36.32 | 51.14 | 25.23 | 6.15 |
| Ties-Merging [84] | 76.58 | 67.22 | 78.61 | 40.74 | 10.54 | 52.69 | 19.90 | 19.13 | 3.91 | 1.50 |
| AdaMerging [85] | 99.13 | 93.38 | 84.19 | 59.90 | 25.70 | 64.09 | 48.66 | 66.55 | 38.54 | 7.94 |
| **EMR-Merging (Ours)** | **100.00** | **98.48** | **89.09** | **95.98** | **82.33** | **76.19** | **74.12** | **87.70** | **87.11** | **96.07** |

| Average Acc | Individual | Weight Averaging | RegMean [33] | Task Arithmetic [30] | Ties-Merging [84] | AdaMerging [85] | **EMR-Merging (Ours)** |
|---|---|---|---|---|---|---|---|
| Acc | 93.02 | 42.52 | 68.14 | 48.89 | 37.53 | 60.25 | **89.54** |

compared to existing merging methods, respectively 7.6% and 2.7%. Notably, EMR-MERGING requires no additional training, tuning, or any dataset accessibility while outperforming AdaMerging and Ties-Merging, which require additional training or careful hyper-parameter tuning using datasets. Under this setting, EMR-MERGING performs very close to or even better than traditional MTL, which is normally considered as a reference upper bound for model merging work [85]. For visualized comparison, we provide some visualization results of different merging methods using t-SNE and Grad-CAM in Fig. 5. It can be seen that among all the merging methods, the visualization results of EMR-MERGING are the closest to individual models, which corresponds to quantitative results. Please check Appendix E for more visualization results.

#### 4.1.2 Merging 30 ViTs.

**Settings.** To further explore the performance of EMR-MERGING, we establish a new benchmark on merging vision models, expanding the number of task-specific models from eight to **30**. We employ ViT-B/16 [22] pre-trained on ImageNet-21k [18] as the pre-trained model. The performance is evaluated by image classification datasets including MNIST [38], CIFAR-10 [36], Vegetables [1], Food-101 [6], Kvasir-v2 [53], Cars [35], Intel Images [4], EuroSAT [27], Weather [82], Cats and dogs [15], MangoLeafBD [2], Beans [37], CIFAR-100 [36], GTSRB [65], SVHN [91], Dogs [34], Fashion MNIST [81], Oxford-IIIT-Pet [50], Landscape Recognition [17], Flowers Recognition [45], STL-10 [12], CUB-200-2011 [73], EMNIST [13], DTD [11], RESISC45 [10], SUN397 [83], Kenyan-Food13 [32], Animal-10N [64], Garbage Classification [8], and Fruits-360 [47], covering tasks from common food classification to disease detection. All of them are evaluated by accuracy.

**Results.** The experimental results are shown in Tab. 4. It can be clearly seen that under this challenging setting of merging **30** models, all the existing methods show significant performance drops compared to individual models. Even RegMean, which performs best among existing methods, still exhibits a performance decay of nearly 25%. However, EMR-MERGING can reduce this value

Table 5: Results of merging RoBERTa models on eight datasets from GLUE benchmark.

| Methods | Single-Sentence Tasks | | Similarity and Paraphrase Tasks | | | Inference Tasks | | |
|---|---|---|---|---|---|---|---|---|
| | **CoLA** | **SST2** | **MRPC** | **STSB** | **QQP** | **MNLI** | **QNLI** | **RTE** |
| Individual | 0.6018 | 0.9404 | 0.8922 | 0.9063 | 0.9141 | 0.8720 | 0.9271 | 0.7906 |
| Weight Averaging | 0.1396 | 0.6411 | 0.6936 | 0.3184 | 0.7536 | 0.4219 | 0.587 | 0.5523 |
| RegMean [33] | 0.3667 | 0.906 | 0.7574 | 0.6268 | 0.8355 | 0.7002 | 0.8235 | 0.5848 |
| Task Arithmetic [30] | 0.1878 | 0.8589 | 0.7990 | 0.7403 | 0.8378 | 0.5908 | 0.6967 | 0.6209 |
| Ties-Merging [84] | 0.2048 | 0.8440 | 0.8113 | 0.5819 | 0.8570 | 0.6465 | 0.7481 | 0.4296 |
| **EMR-MERGING (Ours)** | **0.3996** | **0.9335** | **0.8627** | **0.8277** | **0.8972** | **0.8545** | **0.8957** | **0.7437** |

Table 6: Multi-task performance when merging GPT-2 models on seven text classification tasks.

| Method | CoLA | MNLI | MRPC | QNLI | QQP | RTE | SST-2 | Avg. |
|---|---|---|---|---|---|---|---|---|
| Indivudual | 76.8 | 82.1 | 80.4 | 88.3 | 89.6 | 65.3 | 91.2 | 82.0 |
| Weight Averaging | 55.0 | 55.1 | 51.0 | 57.6 | 76.7 | 44.8 | 52.5 | 56.1 |
| Fisher Merging [46] | 54.8 | 58.0 | 39.5 | 63.3 | 81.5 | 49.1 | 64.7 | 58.7 |
| RegMean [33] | 61.7 | 70.4 | 65.4 | 69.7 | 78.8 | 56.0 | 79.7 | 68.8 |
| Task Arithmetic [30] | 68.7 | 68.6 | 69.6 | 70.5 | 81.8 | 47.3 | 83.6 | 70.0 |
| Ties-Merging [84] | 68.4 | 71.4 | 68.4 | 69.6 | 82.4 | 47.7 | 81.8 | 70.0 |
| **EMR-MERGING (Ours)** | **72.8** | **81.1** | **79.2** | **84.8** | **88.1** | **66.5** | **90.3** | **80.4** |

to 3.48%. This shows that the proposed method maintains the performance comparable to individual models when merging vision models even if the number of tasks increases.

## 4.2 Merging language models

### 4.2.1 Merging fully finetuned RoBERTa models

**Settings.** We partially follow the setting from DARE [90]. However, instead of merging two or three models at a time, we merge all eight models finetuned on each task. RoBERTa-base [43] model is selected as the pre-trained model. The performance of each method is evaluated by eight tasks from GLUE [74] benchmark, respectively CoLA [76], SST-2 [63], MRPC [21], STS-B [9], QQP [31], MNLI [78], QNLI [56], and RTE [24]. Among them, CoLA is evaluated by the Matthews correlation coefficient, STS-B is evaluated by the average value of Pearson and Spearman correlation coefficients, and the rest tasks are evaluated by accuracy.

**Results.** The experimental results are shown in Tab. 5. It can be seen that EMR-MERGING outperforms all the existing methods on every task, verifying the applicability of the proposed method to language models. Note that the reported results of Ties-Merging, Task Arithmetic, and RegMean are the best among multiple hyper-parameter settings. Please check Appendix D.4 for more detailed information. It should also be noted that we find that under our setting of merging multiple models, DARE may not help improve the performance. Similar results were also reported by [25]. This may be due to DARE's random dropping strategy can no longer resolve conflicts among task vectors under the setting of merging multiple models. Please check Appendix D.3 for DARE's experimental results.

### 4.2.2 Merging fully finetuned GPT-2 models

**Settings.** We follow the setting from FusionBench [68], a benchmark for model merging. We merge GPT-2 [55] models on seven tasks from GLUE [74], each with a different head for classification. Under this setting, each task is evaluated by accuracy.

**Results.** The experimental results are shown in Tab. 6. EMR-MERGING outperforms all the merging methods by over 10% and decreases the performance degradation caused by model merging from 12% to 1.6%. This validates the applicability of EMR-MERGING to fully finetuned GPT2-scale language models.

### 4.2.3 Merging PEFT models

**Settings.** We follow the setting from Ties-Merging [84]. (IA)$^3$ [42] is a PEFT method that uses learned vectors to scale the base model activations. We use T0-3B [60] as the base model and merge (IA)$^3$ modules. The performance is evaluated using eleven datasets, including RTE [24], CB [16],

Table 7: Results of merging $(IA)^3$ models on eleven NLP tasks.

| Methods | Validation | RTE | CB | Winogrande | WiC | WSC | COPA | H-SWAG | Story Cloze | ANLI-R1 | ANLI-R2 | ANLI-R3 | Avg Acc |
|---|---|---|---|---|---|---|---|---|---|---|---|---|---|
| Individual | - | 82.7 | 95.8 | 75.1 | 71.7 | 65.3 | 85.3 | 44.4 | 94.9 | 70.2 | 46.5 | 53 | 71.4 |
| Traditional MTL | - | 88.6 | 95.8 | 75.5 | 61.1 | 80.6 | 94.1 | 42.3 | 97.6 | 70.5 | 49.8 | 47.7 | 73.1 |
| Fisher Merging [46] | ✓ | 83.3 | 83.3 | 56.7 | 54.2 | 58.3 | 83.1 | 42.2 | 94.1 | 45.9 | 41.0 | 42.2 | 62.2 |
| RegMean [33] | ✓ | 81.2 | 58.3 | 53.8 | 55.2 | 53.5 | 80.9 | 40.1 | 92.5 | 43.3 | 39.2 | 40.2 | 58 |
| Task Arithmetic [30] | ✓ | 74.1 | 83.3 | 62.8 | 49.1 | 49.3 | 87.5 | 41.5 | 95.3 | 60.8 | 49.4 | 50.0 | 63.9 |
| Ties-Merging [84] | ✓ | 78.0 | 83.3 | 67.9 | 57.6 | 59.7 | 81.7 | 42.8 | 90.3 | 66.9 | 51.3 | 51.1 | 66.4 |
| Weight Averaging | × | 81.2 | 58.3 | 53.8 | 55.2 | 53.5 | 80.9 | 40.1 | 92.5 | 43.3 | 39.2 | 40.2 | 58 |
| Task Arithmetic [30] | × | 76.5 | 79.2 | 57.7 | 51.6 | 51.4 | 66.2 | 31.4 | 81.5 | 59.8 | 47.5 | 48.2 | 59.2 |
| Ties-Merging [84] | × | 81.2 | 87.5 | 60.8 | 59.9 | 58.3 | 80.2 | 42.6 | 91.1 | 58.1 | 46.5 | 47.4 | 64.9 |
| **EMR-Merging (Ours)** | × | **81.8** | **87.5** | **66.6** | **56.1** | **65.3** | **82.4** | **44.7** | **93.6** | **65.7** | **43.8** | **50.8** | **67.1** |

Table 8: Results of merging multi-modal BEiT3 models on five vision-language tasks.

| Methods | Task Metric | COCO-Retrieval Accuracy(↑) | COCO-Captioning | | | | ImageNet-1k Classification Accuracy(↑) | NLVR2 Accuracy(↑) | VQAv2 Accuracy(↑) |
|---|---|---|---|---|---|---|---|---|---|
| | | | BLEU4(↑) | CIDEr(↑) | METEOR(↑) | ROUGE-L(↑) | | | |
| Individual | | 0.8456 | 0.394 | 1.337 | 0.311 | 0.601 | 0.8537 | 0.7765 | 0.8439 |
| Weight Averaging | | 0.1893 | 0.031 | 0.001 | 0.115 | 0.159 | 0.6771 | 0.2800 | 0.6285 |
| Task Arithmetic [30] | | 0.3177 | 0.033 | 0.000 | 0.118 | 0.176 | 0.7081 | 0.3809 | 0.6933 |
| Ties-Merging [84] | | 0.3929 | 0.029 | 0.001 | 0.108 | 0.167 | 0.6978 | 0.3206 | 0.6717 |
| **EMR-Merging**(Ours) | | **0.7946** | **0.289** | **1.060** | **0.272** | **0.534** | **0.7742** | **0.7475** | **0.7211** |

Winogrande [59], WiC [52], WSC [39], COPA [58], H-SWAG [92], Story Cloze [62], and ANLI [48] from R1 to R3. All the datasets are evaluated by accuracy.

**Results.** The experimental results are shown in Tab. 7. EMR-MERGING outperforms all the merging methods. Compared to methods without validation, EMR-MERGING improves the average accuracy on each task by $2.2\%$. When compared to methods that require validation data to tune hyper-parameters or compute matrices for weighted merging, EMR-MERGING still improves the average performance by $0.7\%$, validating the applicability of our method to PEFT models.

### 4.3 Merging multi-modal models

**Settings.** We merge BEiT3-base [75] models finetuned on five datasets from different kinds of tasks, respectively ImageNet-1k [18] (Image Classification), VQAv2 [26] (Visual Question Answering), NLVR2 [67] (Visual Reasoning), COCO Captioning [41] (Image Captioning), and COCO Retrieval [41] (Image-Text Retrieval). Among them, COCO Captioning is evaluated by BLEU4 [49], CIDEr [72], METEOR [3], and ROUGE-L [40]. The other tasks are evaluated by accuracy.

**Results.** The experimental results are shown in Tab. 8. It can be seen that EMR-MERGING performs best on all the vision-language tasks regardless of which evaluation metric is applied among all the merging methods, validating the effectiveness of EMR-MERGING in merging multi-modal models.

### 4.4 Merging different number of models

In Fig. 6, we compare the number of parameters and performance using individual models, Ties-Merging, and EMR-MERGING when merging different numbers of ViT-B/32 models under the setting of no-validation.

**Number of parameters.** Compared to other merging methods, EMR-MERGING requires a little additional storage for task-specific modulators. However, compared to a single 32-bit model, the additional storage caused by a task-specific 1-bit mask equals a binarized network, whose size is 32 times smaller than a single 32-bit

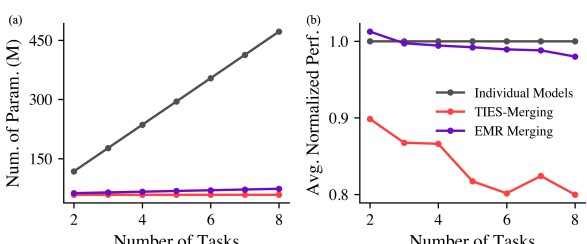

Figure 6: Comparison of the (a) number of parameters and (b) average normalized performance when using individual models, Ties-Merging, and EMR-MERGING.

model [14]. Additionally, the storage required by a task-specific rescaler, which is a single parameter, is negligible. In Fig. 6(a), we compare the number of parameters when merging different numbers of models, and we observe that EMR-MERGING's parameter number is slightly more than Ties-Merging but significantly fewer than individual models.

Table 9: Ablation on the Electing procedure of EMR-MERGING.

| Methods | SUN397 | Cars | RESISC45 | EuroSAT | SVHN | GTSRB | MNIST | DTD | Avg Acc |
|---|---|---|---|---|---|---|---|---|---|
| Task Arithmetic | 63.8 | 62.1 | 72.0 | 77.6 | 74.4 | 65.1 | 94.0 | 52.2 | 70.1 |
| Task Arithmetic w/ M&R | **67.6** | **67.3** | **80.2** | **91.3** | **79.3** | **75.7** | **96.0** | **57.9** | **76.9** [↑ 6.8] |
| Ties-Merging | 64.8 | 62.9 | 74.3 | 78.9 | 83.1 | 71.4 | 97.6 | 56.2 | 73.6 |
| Ties-Merging w/ M&R | **68.8** | **68.9** | **82.2** | **91.6** | **81.4** | **80.0** | 96.6 | **59.3** | **78.6** [↑ 5.0] |
| AdaMerging++ | 66.6 | 68.3 | 82.2 | 94.2 | 89.6 | 89.0 | 98.3 | 60.6 | 81.1 |
| AdaMerging++ w/ M&R | **74.0** | **76.2** | **93.1** | **98.2** | **93.3** | **96.3** | **99.4** | **71.2** | **87.7**[↑ 6.6] |
| EMR-MERGING (Ours) | **75.2** | **72.8** | **93.5** | **99.5** | **96.9** | **98.1** | **99.6** | **74.4** | **88.7** |

Table 10: Ablation on the Masking and Rescaling procedures of EMR-MERGING.

| Methods | SUN397 | Cars | RESISC45 | EuroSAT | SVHN | GTSRB | MNIST | DTD | Avg Acc |
|---|---|---|---|---|---|---|---|---|---|
| Ours (Elect) | 31.7 | 34.7 | 51.8 | 65.9 | 85.7 | 64.0 | 98.2 | 42.2 | 59.3 |
| Ours (Elect & Mask) | 70.7 | 65.9 | 92.2 | 98.7 | 96.9 | 97.6 | 99.6 | 72.3 | 86.8 [↑ 27.5] |
| Ours (Elect & Rescale) | 58.2 | 57.2 | 69.1 | 81.6 | 85.2 | 73.0 | 98.4 | 52.2 | 71.9 [↑ 12.6] |
| Ours (Elect, Mask & Rescale) | **75.2** | **72.8** | **93.5** | **99.5** | **96.9** | **98.1** | **99.6** | **74.4** | **88.7** [↑ 29.4] |

**Performance.** The performance comparison when merging different numbers of models is shown in Fig. 6(b). Compared to Ties-Merging, the performance of EMR-MERGING is higher and decreases more slowly as the task increases. Note that EMR-MERGING outperforms individual models under the 2-task setting. Similar findings are reported by DARE [90]. More details are shown in Appendix D.5.

## 4.5 Ablation Study

We perform ablations on all the components of EMR-MERGING as follows.

**Ablation on Electing procedure.** Tab. 9 shows the results of merging eight ViT-B/32 models when the Electing procedure is replaced by other task vector-based merging methods. The effectiveness of our Electing strategy is verified by outperforming the combination of other merging methods with masking and rescaling. Another interesting finding is that as a post-processing procedure, masking and rescaling can help improve the performance of task vector-based merging methods, respectively 6.8%, 5.0%, and 6.6% for Task Arithmetic, Ties-Merging, and AdaMerging++.

**Ablation on Masking and Rescaling procedures.** Then, we further validate the importance of Masking and Rescaling procedures by disabling either or both of them. The results are shown in Tab. 10. It can be seen that simply electing results in a severe performance drop while adding Masking and Rescaling can improve the performance by 27.5% and 12.6%, respectively. Furthermore, compared to separately applying either of these two procedures, jointly applying Masking and Rescaling leads to greater improvement, up to 29.4%.

## 5 Conclusion

In this paper, we study on tuning-free and high-performance model merging. We first attribute the severe performance degradation of existing merging methods to that a single model can hardly simulate all the models' performance. Then we propose ELECT, MASK & RESCALE-MERGING (EMR-MERGING), which does not require any data access or additional training for tuning. The effectiveness of EMR-MERGING is validated by comprehensive experiments on various classical benchmarks and newly-established benchmarks under vision, NLP, PEFT, and multi-modal settings.

## 6 Acknowledgement

This work is supported by Shanghai Natural Science Foundation (No. 23ZR1402900), National Natural Science Foundation of China (No. 62071127 and No. 62306261), National Key Research and Development Program of China (No. 2022ZD0160101). The computations in this research were performed using the CFFF platform of Fudan University.

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

# Appendix for EMR-MERGING

## A    Algorithm flow of EMR-MERGING

We summarize the procedure of EMR-MERGING in Algorithm 1.

---

**Algorithm 1** EMR-MERGING Procedure

---

**Input:** Finetuned models $W_{1..N}$, pretrained model $W_{pre}$
**Output:** Unified task vector $\tau_{uni}$, task-specific masks $M_{1..N}$, task-specific rescalers $\lambda_{1..N}$
**for** $t$ **in** $1, ..., N$ **do**
   ▷ Create task vectors.
   $\tau_t = W_t - W_{pre}$
**end**
▷ Step 1:  Elect the unified task vector.
$\gamma_{uni} = sgn(\sum_{t=1}^n \tau_t)$
$\epsilon_{uni} = zeros(d)$
**for** $t$ **in** $1, ..., N$ **do**
   **for** $p$ **in** $1, ..., d$ **do**
      **if** $\gamma_{uni}^p \cdot \tau_t^p > 0$ **then**
        $\epsilon_{uni}^p = max\,(\epsilon_{uni}^p, abs\,(\gamma_{uni}^p))$
      **end**
   **end**
**end**
$\tau_{uni} = \gamma_{uni} \odot \epsilon_{uni}.$
**for** $t$ **in** $1, ..., N$ **do**
   ▷ Step 2:  Generate task-specific masks.
   **for** $p$ **in** $1, ..., d$ **do**
      $M_t^p = bool(\tau_t^p \odot \tau_{uni}^p > 0)$
   **end**
   ▷ Step 3:  Generate task-specific rescalers.
   $\lambda_t = \frac{sum(abs(\tau_t))}{sum(abs(M_t \cdot \tau_{uni}))}$
**end**

---

## B    Theoretical analyses

In Section 3, we claimed that the task-specific modulators can lower the distance between the merged model and task-specific models. Here we provide detailed theoretical analyses.

Our goal is to merge model weights $W_{1..N}$ by minimizing the distance between the merged model $W_{uni}$ and each individual models $W_i$, $i \in [1..N]$ **without** using any dataset $[X_i, Y_i]$, where the distance can be calculated by:

$$Dis = \frac{\sum_{i=1}^N \|W_i - W_{uni}\|^2}{N} \tag{6}$$

The premise of merging is that all the models are fine-tuned from the same pre-trained model. Thus, Eq. 6 can be re-written:

$$Dis = \frac{\sum_{i=1}^N \|\tau_i - \tau_{uni}\|^2}{N} \tag{7}$$

where $\tau_i$ refers to the task vector for Task $i$. $\tau_{uni}$ is the merged task vector. We demonstrate the effectiveness of the task-specific modulators by step.

**Analysis 1: Effectiveness of Masks.** Suppose we apply a mask $M_i$ to the unified model $\tau_{uni}$ to disable elements in $\tau_{uni}$ that have the opposite sign of the corresponding elements in $\tau_{uni}$, which can be written as:

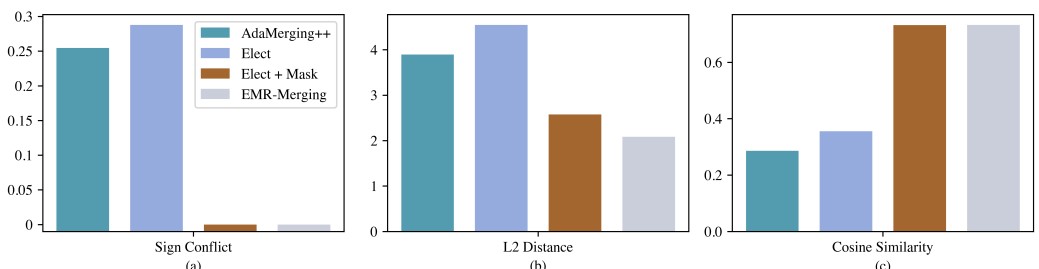

Figure 7: Comparison of (a) sign conflicts, (b) L2 distance, and (c) cosine similarity of model weights obtained by different methods (including AdaMerging++ and each procedure of EMR-MERGING) and task-specific model weights. The detailed configuration is shown in Appendix F.

Table 11: Multi-task performance when merging ViT-B/16 models on eight tasks.

| Methods | SUN397 | Cars | RESISC45 | EuroSAT | SVHN | GTSRB | MNIST | DTD | Avg Acc |
|---|---|---|---|---|---|---|---|---|---|
| Task Arithmetic [30] | 61.1 | 65.9 | 74.0 | 76.2 | 88.0 | 73.9 | 98.4 | 53.0 | 73.8 |
| Ties-Merging [84] | 69.1 | 72.5 | 80.5 | 84.0 | 85.0 | 71.5 | 98.1 | 54.9 | 77.0 |
| AdaMerging [85] | 70.2 | 80.7 | 81.6 | 94.8 | 91.6 | 95.8 | 98.5 | 66.2 | 84.9 |
| AdaMerging++ [85] | 71.8 | 80.8 | 84.1 | 94.3 | 91.9 | 94.5 | 98.7 | 69.8 | 85.7 |
| **EMR-MERGING (Ours)** | **78.6** | **82.6** | **95.5** | **99.2** | **97.6** | **98.8** | **99.6** | **78.3** | **91.3** |

$$M_i = (\tau_i \odot \tau_{uni} > 0) \tag{8}$$

By applying the masks $M_i, i \in [1..N]$, the distance becomes:

$$Dis^M = \frac{\sum_{i=1}^{N} \|\tau_i - M_i \odot \tau_{uni}\|^2}{N} \tag{9}$$

Furthermore, it can be written as:

$$
\begin{aligned}
Dis^M &= \frac{\sum_{i=1}^{N} \|M_i \odot \tau_i - M_i \odot \tau_{uni}\|^2}{N} + \frac{\sum_{i=1}^{N} \|(1 - M_i) \odot \tau_i\|^2}{N} \\
&= \frac{\sum_{i=1}^{N} \|M_i \odot (abs(\tau_i) - abs(\tau_{uni}))\|^2}{N} + \frac{\sum_{i=1}^{N} \|(1 - M_i) \odot abs(\tau_i)\|^2}{N}
\end{aligned}
\tag{10}
$$

where $abs(\cdot)$ returns the absolute value of each element in the input. For ease of comparison, the distance without applying $M_i$ can be formulated as:

$$
\begin{aligned}
Dis &= \frac{\sum_{i=1}^{N} \|M_i \odot (abs(\tau_i) - abs(\tau_{uni}))\|^2}{N} + \frac{\sum_{i=1}^{N} \|(1 - M_i) \odot (abs(\tau_i) + abs(\tau_{uni}))\|^2}{N} \\
&= Dis^M + \frac{\sum_{i=1}^{N} \|(1 - M_i) \odot abs(\tau_{uni})\|^2}{N}
\end{aligned}
\tag{11}
$$

Thus, we demonstrate that $Dis^M \le Dis$, indicating applying task-specific masks can reduce the distance between the merged model and individual models, thus showing effectiveness.

**Analysis 2: Effectiveness of Rescalers.** Suppose we apply a rescaler $\lambda_i > 0$ to the masked unified task vector $M_i \cdot \tau_{uni}$, the distance becomes:

$$
\begin{aligned}
Dis^{M,\lambda} &= \frac{\sum_{i=1}^{N} \|\tau_i - \lambda_i \cdot M_i \odot \tau_{uni}\|^2}{N} \\
&= \frac{\sum_{i=1}^{N} \|abs(\tau_i) - \lambda_i \cdot abs(M_i \odot \tau_{uni})\|^2}{N}
\end{aligned}
\tag{12}
$$

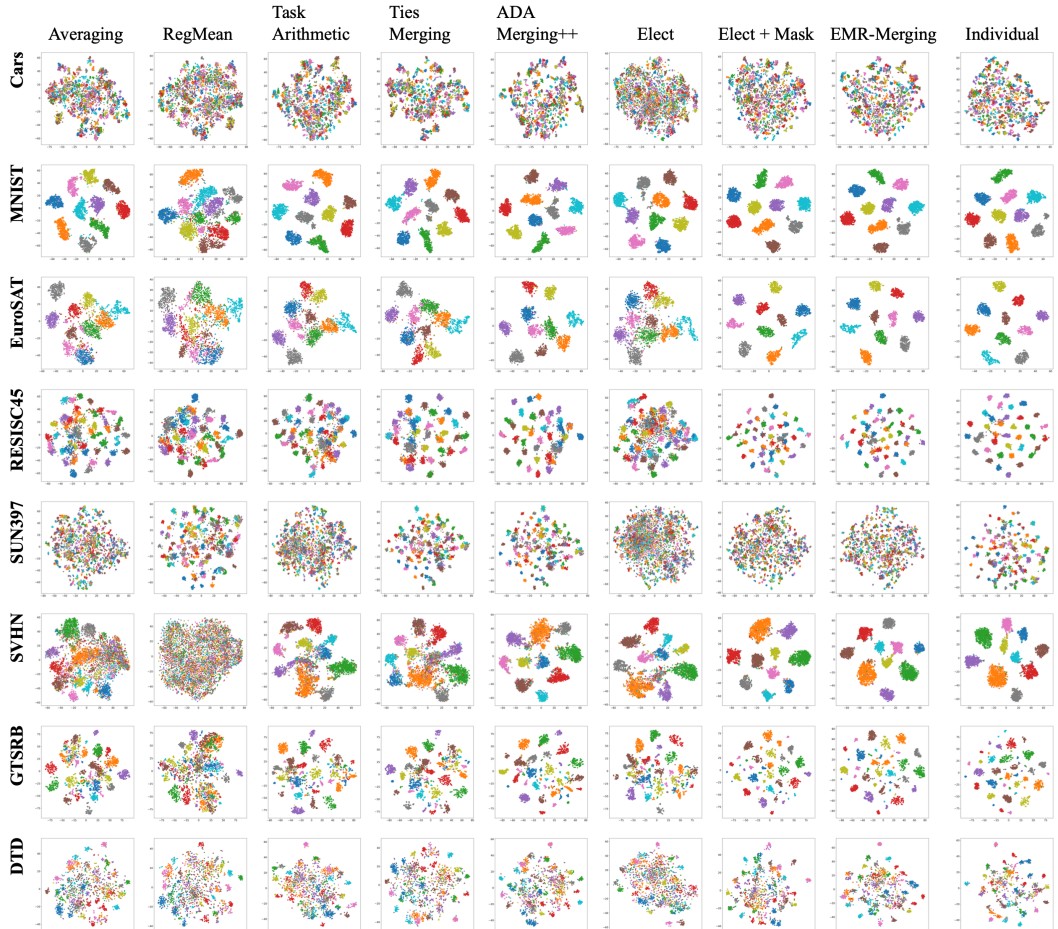

Figure 8: t-SNE visualization results of different merging methods.

Table 12: Multi-task performance when merging ViT-B/32 models on 9 vision tasks (ImageNet-1K added).

| Methods | SUN397 | Cars | RESISC45 | EuroSAT | SVHN | GTSRB | MNIST | DTD | ImageNet-1K | Avg Acc |
|---|---|---|---|---|---|---|---|---|---|---|
| Individual | 75.3 | 77.7 | 96.1 | 99.7 | 97.5 | 98.7 | 99.7 | 79.4 | 82.0 | 89.6 |
| Weight Averaging | 61.8 | 56.4 | 65.9 | 66.2 | 62.7 | 44.5 | 81.8 | 49.0 | 61.5 | 61.1 |
| Task Arithmetic [30] | 51.8 | 30.9 | 55.8 | 64.3 | 69.0 | 42.2 | 92.7 | 46.8 | 66.6 | 57.8 |
| Ties-Merging [84] | 53.3 | 34.1 | 57.0 | 55.8 | 72.3 | 43.2 | 90.5 | 46.5 | 68.9 | 58.0 |
| **EMR-MERGING** (Ours) | **77.0** | **75.2** | **92.9** | **92.7** | **79.7** | **90.2** | **97.6** | **76.2** | **79.8** | **84.6** |

To minimize the distance in Eq. 12, we set the first derivative of $Dis^\lambda$ with respect to $\lambda_i$ to 0, thus $\lambda_i$ can be calculated by:

$$\lambda_i = \frac{sum(abs(\tau_i))}{sum(abs(M_i \odot \tau_{uni}))} \tag{13}$$

which exactly matches our setting of $\lambda_i$. This indicates that our setting of rescalers $\lambda_i$ can minimize the distance between the merged model and individual models, which is: $Dis^{M,\lambda} \leq Dis^M$, thus showing effectiveness.

It is also reflected in Fig. 7 that after Masking and Rescaling, the sign conflicts and L2 distance between the merged model and task-specific models are reduced and the cosine similarity can is improved.

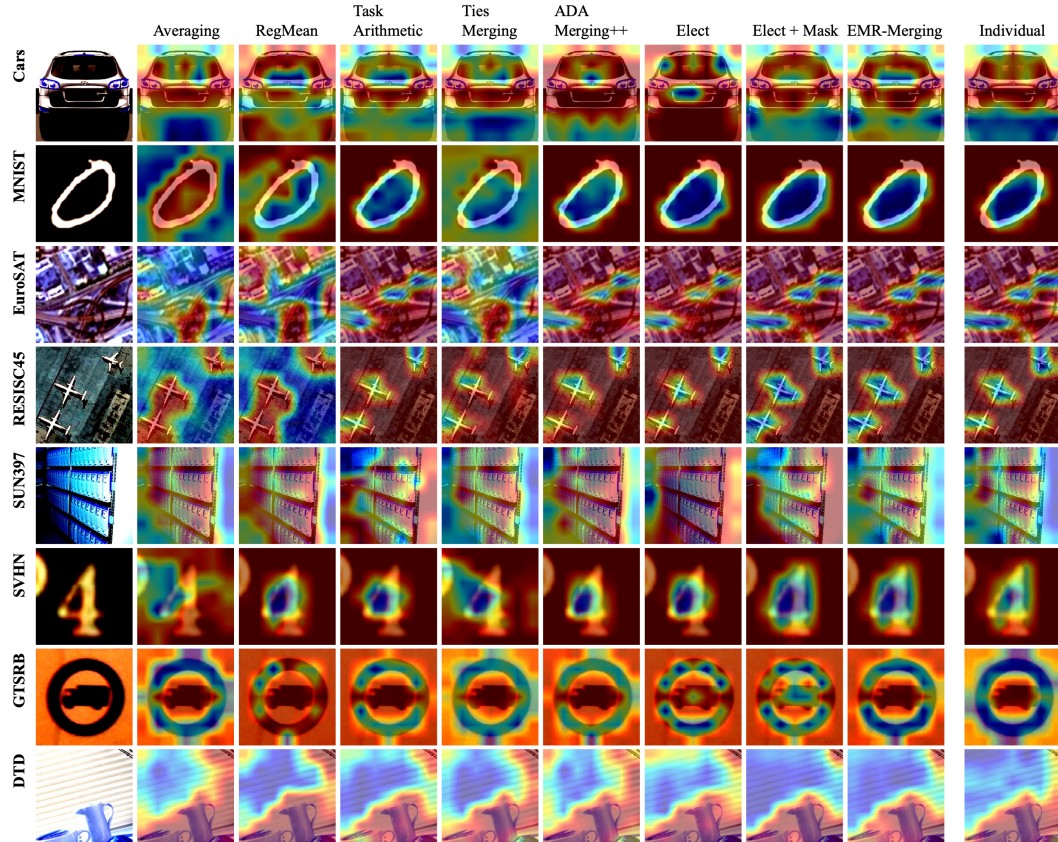

Figure 9: Grad-CAM visualization results of different merging methods.

## C  Baseline Methods

- **Individual Models** refer to task-specific models before merging.

- **Traditional MTL** uses datasets from all the tasks to train a single model jointly.

- **Weight Averaging** element-wisely averages all the model weights. Its effectiveness when applied to fine-tuned model weights from the same pre-training has been verified [80, 57, 33].

- **Fisher Merging** [46] uses Fisher information matrices [23] to calculate the importance of each parameter and weighted merges them based on their importance.

- **RegMean** [33] weighted merges models based on a closed-form solution to the merging problem. When merging $K$ linear model weights $W_i$, where $f_i(x) = W_i^T x$, $i = 1..K$, the merging problem can be formulated as: $\min_{W} \sum_{i=1}^{K} \|W^T X_i - W_i^T X_i\|^2$, where $W$ is the merged model weights, and $X_i$ denotes the input of $i^{th}$ model. The closed-form solution to the problem is: $W = (\sum_{i=1}^{K} X_i^T X_i)^{-1} (\sum_{i=1}^{K} X_i^T X_i W_i)$. Inner-product matrices need to be computed before merging.

- **Task Arithmetic** [30] defines task vectors as the difference between finetuned model weights and the pre-trained model weights. Suppose a model $\theta_i$ is finetuned from a pre-trained model $\theta_{pre}$, the task vector is $\tau_i = \theta_i - \theta_{pre}$. When merging $\theta_{1..K}$, the merged model is $\theta_M = \lambda \sum_{i=1}^{K} \tau_i + \theta_{pre}$, where $\lambda$ is the merging coefficient.

- **Ties-Merging** [84] (Trim, Elect Sign & Merge) believes that the conflicts among the task vectors severely effect the merged model's performance. Ties-Merging solves this problem by eliminating redundant parameters and resolving symbol conflicts.

Table 13: Performance of RegMean and Task Arithmetic when pre-processed using DARE [90].

| Methods | Single-Sentence Tasks | | Similarity and Paraphrase Tasks | | | Inference Tasks | | |
| --- | --- | --- | --- | --- | --- | --- | --- | --- |
| | **CoLA** | **SST2** | **MRPC** | **STSB** | **QQP** | **MNLI** | **QNLI** | **RTE** |
| Individual | 0.6018 | 0.9404 | 0.8922 | 0.9063 | 0.9141 | 0.8720 | 0.9271 | 0.7906 |
| **EMR-MERGING** (Ours) | 0.3996 | 0.9335 | 0.8627 | 0.8277 | 0.8972 | 0.8545 | 0.8957 | 0.7437 |
| RegMean [33] | 0.3667 | 0.906 | 0.7574 | 0.6268 | 0.8355 | 0.7002 | 0.8235 | 0.5848 |
| w/ DARE (drop 10%) | 0.5046 | 0.5298 | 0.3603 | 0.1533 | 0.4955 | 0.3245 | 0.4924 | 0.4477 |
| w/ DARE (drop 30%) | 0.4535 | 0.6135 | 0.3186 | 0.0471 | 0.4219 | 0.3325 | 0.505 | 0.5126 |
| w/ DARE (drop 50%) | 0.2758 | 0.5138 | 0.3211 | -0.0965 | 0.3685 | 0.3338 | 0.508 | 0.5235 |
| w/ DARE (drop 70%) | 0 | 0.4908 | 0.3162 | 0.0021 | 0.3682 | 0.3184 | 0.5056 | 0.4838 |
| w/ DARE (drop 90%) | 0 | 0.4908 | 0.3162 | -0.0776 | 0.3682 | 0.3187 | 0.5158 | 0.4910 |
| Task Arithmetic [30] | 0.1878 | 0.8589 | 0.7990 | 0.7403 | 0.8378 | 0.5908 | 0.6967 | 0.6209 |
| w/ DARE (drop 10%) | 0.2424 | 0.8509 | 0.7966 | 0.7234 | 0.8382 | 0.5869 | 0.7368 | 0.6101 |
| w/ DARE (drop 30%) | 0.3040 | 0.8452 | 0.7941 | 0.6311 | 0.8333 | 0.5515 | 0.786 | 0.6137 |
| w/ DARE (drop 50%) | 0.2451 | 0.8188 | 0.7990 | 0.4262 | 0.8099 | 0.4591 | 0.7269 | 0.6029 |
| w/ DARE (drop 70%) | 0 | 0.7225 | 0.6373 | 0.1353 | 0.7321 | 0.3453 | 0.6495 | 0.5162 |
| w/ DARE (drop 90%) | 0 | 0.4908 | 0.3162 | 0.0422 | 0.3682 | 0.3185 | 0.5114 | 0.4729 |
| Ties-Merging [84] | 0.2048 | 0.8440 | 0.8113 | 0.5819 | 0.8570 | 0.6465 | 0.7481 | 0.4296 |
| w/ DARE (drop 30%) | 0 | 0.5103 | 0.3382 | -0.0024 | 0.3961 | 0.3238 | 0.5277 | 0.4838 |
| w/ DARE (drop 50%) | 0.0464 | 0.6021 | 0.5343 | 0.0192 | 0.6846 | 0.3410 | 0.5841 | 0.4982 |
| w/ DARE (drop 70%) | 0.1342 | 0.7833 | 0.7672 | 0.1667 | 0.8180 | 0.4172 | 0.691 | 0.5271 |
| w/ DARE (drop 90%) | 0.2618 | 0.8383 | 0.8039 | 0.6082 | 0.8336 | 0.5551 | 0.7692 | 0.5235 |

- **AdaMerging** [85] uses an unsupervised method to learn the merging coefficients for each task vector (Task-wise AdaMerging) or each layer (Layer-wise AdaMerging). AdaMerging++ is realized by adopting Ties-Merging [84] before learning the merging coefficients.

- **DARE** [90] (Drop and Rescale) validates the extremely redundant properties of language models. As a pre-processing technique, DARE randomly drops most (90% or even 99%) delta parameters (task vectors) before merging to potentially mitigate the interference of parameters among models.

# D   More experimental results

## D.1   Merging ViT-B/16 models on 8 tasks

We follow the settings in Section 4.1.1 and merge ViT-B/16 models. Tab. 11 shows the accuracy of merging ViT-B/16 models on eight vision tasks. The proposed EMR-MERGING brings about 5.6% performance improvement compared to Adamerging++ [85], further demonstrating the effectiveness of EMR-MERGING.

## D.2   Merging ViT-B/32 models on 9 tasks (ImageNet-1K added)

To further explore the performance of EMR-MERGING, we follow the settings in Section 4.1.1 and add one more task, ImageNet-1K [18]. We merge models on these nine tasks using different merging methods. The results are shown in Tab. 12 and EMR-Merging shows a much more significant improvement compared to existing merging methods (up to 20%).

## D.3   DARE's experimental results and causes

DARE's experimental results when combined with RegMean and Task Arithmetic are shown in Tab. 13. It can be seen that when applied to merge eight models, DARE works on a few tasks under low dropping rate settings but it generally fails. We attribute its failure to the random dropping strategy's unapplicability to merging multiple models. Under the setting of merging two or three models, randomly dropping most parameters in task vectors can significantly reduce interference but conflicts are a lot more difficult to avoid when merging multiple models.

Table 14: Performance of Task Arithmetic [30], Ties-Merging [84], Ties-Merging [84] w/ DARE [90], and RegMean [33] under different hyper-parameter settings. $\lambda$ for task vector-based methods is the merging coefficient. $P$ is the drop rate for DARE. $a$ is the non-diagonal multiplier for RegMean.

| Methods | Single-Sentence Tasks | | Similarity and Paraphrase Tasks | | | Inference Tasks | | |
|---|---|---|---|---|---|---|---|---|
| | CoLA | SST2 | MRPC | STSB | QQP | MNLI | QNLI | RTE |
| Individual | 0.6018 | 0.9404 | 0.8922 | 0.9063 | 0.9141 | 0.872 | 0.9271 | 0.7906 |
| **EMR-MERGING (Ours)** | | | | | | | | |
| | 0.3996 | 0.9335 | 0.8627 | 0.8277 | 0.8972 | 0.8545 | 0.8957 | 0.7437 |
| **Task Arithmetic** | | | | | | | | |
| $\lambda = 0.1$ | 0.0464 | 0.742 | 0.6691 | 0.2344 | 0.771 | 0.3567 | 0.6919 | 0.556 |
| $\lambda = 0.3$ | 0.1878 | 0.8589 | 0.799 | 0.7403 | 0.8378 | 0.5908 | 0.6967 | 0.6209 |
| $\lambda = 0.5$ | -0.0089 | 0.7913 | 0.7794 | 0.5686 | 0.8271 | 0.4631 | 0.5387 | 0.4693 |
| $\lambda = 0.7$ | -0.0079 | 0.6525 | 0.7819 | 0.1292 | 0.8146 | 0.3949 | 0.5279 | 0.5054 |
| $\lambda = 0.9$ | -0.0207 | 0.7202 | 0.4167 | -0.1283 | 0.8012 | 0.2913 | 0.5294 | 0.5162 |
| $\lambda = 1.0$ | 0 | 0.5619 | 0.3554 | -0.2496 | 0.7939 | 0.259 | 0.5338 | 0.5162 |
| **Ties-Merging** | | | | | | | | |
| $\lambda = 0.1$ | 0 | 0.4908 | 0.3162 | 0.0214 | 0.3682 | 0.3186 | 0.5105 | 0.4729 |
| $\lambda = 0.3$ | 0 | 0.5631 | 0.5049 | -0.0074 | 0.4696 | 0.35 | 0.5649 | 0.4621 |
| $\lambda = 0.5$ | 0.2232 | 0.7592 | 0.7696 | 0.1149 | 0.827 | 0.4486 | 0.6939 | 0.4368 |
| $\lambda = 0.7$ | 0.2507 | 0.8291 | 0.7917 | 0.3774 | 0.8488 | 0.5858 | 0.7507 | 0.4188 |
| $\lambda = 0.9$ | 0.2048 | 0.844 | 0.8113 | 0.5819 | 0.857 | 0.6465 | 0.7481 | 0.4296 |
| $\lambda = 1.0$ | 0.1712 | 0.8406 | 0.799 | 0.6444 | 0.859 | 0.6409 | 0.7069 | 0.426 |
| **Ties-Merging w/ DARE** | | | | | | | | |
| $\lambda = 0.2, P = 0.3$ | 0 | 0.4920 | 0.3162 | 0.0053 | 0.3682 | 0.3186 | 0.5131 | 0.4477 |
| $\lambda = 0.2, P = 0.5$ | 0 | 0.0043 | 0.3162 | 0.0036 | 0.3690 | 0.3202 | 0.5226 | 0.4946 |
| $\lambda = 0.2, P = 0.7$ | 0.0464 | 0.6388 | 0.5735 | 0.0301 | 0.0047 | 0.3383 | 0.5984 | 0.5090 |
| $\lambda = 0.2, P = 0.9$ | 0.2402 | 0.8165 | 0.7843 | 0.2696 | 0.8112 | 0.4384 | 0.7223 | 0.5415 |
| $\lambda = 0.3, P = 0.3$ | 0 | 0.5103 | 0.3382 | -0.0024 | 0.3961 | 0.3238 | 0.5277 | 0.4838 |
| $\lambda = 0.3, P = 0.5$ | 0.0464 | 0.6021 | 0.5343 | 0.0192 | 0.6846 | 0.3410 | 0.5841 | 0.4982 |
| $\lambda = 0.3, P = 0.7$ | 0.1342 | 0.7833 | 0.7672 | 0.1667 | 0.8180 | 0.4172 | 0.691 | 0.5271 |
| $\lambda = 0.3, P = 0.9$ | 0.2618 | 0.8383 | 0.8039 | 0.6082 | 0.8336 | 0.5551 | 0.7692 | 0.5235 |
| $\lambda = 0.4, P = 0.3$ | 0.0656 | 0.6216 | 0.5588 | 0.0192 | 0.7301 | 0.3461 | 0.5891 | 0.5162 |
| $\lambda = 0.4, P = 0.5$ | 0.1172 | 0.7374 | 0.7451 | 0.1045 | 0.8157 | 0.3913 | 0.6667 | 0.5126 |
| $\lambda = 0.4, P = 0.7$ | 0.2440 | 0.8234 | 0.7843 | 0.3955 | 0.8371 | 0.5496 | 0.7216 | 0.4838 |
| $\lambda = 0.4, P = 0.9$ | 0.1380 | 0.8440 | 0.8064 | 0.7044 | 0.8365 | 0.5835 | 0.6529 | 0.5054 |
| **RegMean** | | | | | | | | |
| $a = 0.7$ | 0.3005 | 0.9037 | 0.7525 | 0.6349 | 0.8322 | 0.6794 | 0.8157 | 0.5632 |
| $a = 0.8$ | 0.3346 | 0.9014 | 0.7549 | 0.6375 | 0.8339 | 0.6841 | 0.8173 | 0.5704 |
| $a = 0.9$ | 0.3445 | 0.9048 | 0.7525 | 0.6362 | 0.8361 | 0.6918 | 0.821 | 0.5632 |
| $a = 1.0$ | 0.3667 | 0.906 | 0.7574 | 0.6268 | 0.8355 | 0.7002 | 0.8235 | 0.5848 |

## D.4 Results under different hyper-paramerter settings

In Section 4.2.1, we presented the best performance of Ties-Merging, Task Arithmetic, and RegMean among multiple hyper-parameter settings. Here we present more experimental results of Ties-Merging, Task Arithmetic, and RegMean under different hyper-parameter settings in Tab. 14.

## D.5 Detailed information for merging different number of models

In Section 4.4, we showed partial results of merging different number of ViT-B/32 models by Fig. 6. Here we provide quantified and task-specific performance results in Tab. 15.

Table 15: Merging different number of ViT-B/32 models.

| Methods | SUN397 | Cars | RESISC45 | EuroSAT | SVHN | GTSRB | MNIST | DTD | Avg Acc |
|---|---|---|---|---|---|---|---|---|---|
| **Individual** | | | | | | | | | |
| 2 Tasks | 75.3 | 77.7 | - | - | - | - | - | - | 76.5 |
| 3 Tasks | 75.3 | 77.7 | 96.1 | - | - | - | - | - | 83.0 |
| 4 Tasks | 75.3 | 77.7 | 96.1 | 99.7 | - | - | - | - | 87.2 |
| 5 Tasks | 75.3 | 77.7 | 96.1 | 99.7 | 97.5 | - | - | - | 89.3 |
| 6 Tasks | 75.3 | 77.7 | 96.1 | 99.7 | 97.5 | 98.7 | - | - | 90.8 |
| 7 Tasks | 75.3 | 77.7 | 96.1 | 99.7 | 97.5 | 98.7 | 99.7 | - | 92.1 |
| 8 Tasks | 75.3 | 77.7 | 96.1 | 99.7 | 97.5 | 98.7 | 99.7 | 79.4 | 90.5 |
| **Ties-Merging** | | | | | | | | | |
| 2 Tasks | 69.2 | 68.2 | - | - | - | - | - | - | 68.7 |
| 3 Tasks | 69.2 | 68.0 | 78.9 | - | - | - | - | - | 72.0 |
| 4 Tasks | 68.9 | 67.9 | 79.4 | 86.0 | - | - | - | - | 75.5 |
| 5 Tasks | 68.6 | 67.1 | 79.0 | 83.5 | 66.6 | - | - | - | 73.0 |
| 6 Tasks | 68.0 | 66.4 | 77.9 | 80.1 | 74.4 | 69.9 | - | - | 72.8 |
| 7 Tasks | 66.6 | 65.7 | 75.7 | 76.7 | 81.0 | 69.2 | 96.4 | - | 75.9 |
| 8 Tasks | 64.8 | 62.9 | 74.3 | 78.9 | 83.1 | 71.4 | 97.6 | 56.2 | 72.4 |
| **EMR-MERGING (Ours)** | | | | | | | | | |
| 2 Tasks | 78.9 | 76.1 | - | - | - | - | - | - | 77.5 |
| 3 Tasks | 77.9 | 75.2 | 95.3 | - | - | - | - | - | 82.8 |
| 4 Tasks | 77.4 | 74.9 | 94.8 | 99.7 | - | - | - | - | 86.7 |
| 5 Tasks | 77.2 | 74.2 | 94.7 | 99.7 | 97.1 | - | - | - | 88.6 |
| 6 Tasks | 76.4 | 73.4 | 94.2 | 99.7 | 97.0 | 98.5 | - | - | 89.9 |
| 7 Tasks | 75.8 | 73.3 | 93.6 | 99.6 | 96.9 | 98.2 | 99.6 | - | 91.0 |
| 8 Tasks | 75.2 | 72.8 | 93.5 | 99.5 | 96.9 | 98.1 | 99.6 | 74.4 | 88.7 |

Table 16: Sparsity (ratio of non-zero items) of the masks and the values of the rescalers when merging ViTs on 8 vision tasks and RoBERTa models on 8 language tasks.

| Sparsity | SUN397 | Cars | RESISC45 | EuroSAT | SVHN | GTSRB | MNIST | DTD |
|---|---|---|---|---|---|---|---|---|
| ViT-B/32 | 0.7194 | 0.7121 | 0.7106 | 0.6994 | 0.7195 | 0.7062 | 0.7132 | 0.7058 |
| ViT-L/14 | 0.6832 | 0.6699 | 0.6734 | 0.6579 | 0.6748 | 0.6444 | 0.6614 | 0.6620 |

| Rescalers | SUN397 | Cars | RESISC45 | EuroSAT | SVHN | GTSRB | MNIST | DTD |
|---|---|---|---|---|---|---|---|---|
| ViT-B/32 | 0.7489 | 0.7635 | 0.7489 | 0.7476 | 0.7962 | 0.7652 | 0.7981 | 0.7624 |
| ViT-L/14 | 0.7656 | 0.7652 | 0.7537 | 0.7384 | 0.7874 | 0.7313 | 0.7763 | 0.7638 |

| Sparsity | CoLA | SST2 | MRPC | STSB | QQP | MNLI | QNLI | RTE |
|---|---|---|---|---|---|---|---|---|
| RoBERTa | 0.6264 | 0.6547 | 0.6498 | 0.6150 | 0.7620 | 0.7739 | 0.6243 | 0.5979 |

| Rescalers | CoLA | SST2 | MRPC | STSB | QQP | MNLI | QNLI | RTE |
|---|---|---|---|---|---|---|---|---|
| RoBERTa | 0.2458 | 0.4698 | 0.5033 | 0.2078 | 0.8891 | 0.8987 | 0.4683 | 0.1466 |

### D.6 Sparsity of masks and values of rescalers.

We show the sparsity of the masks and the values of the rescalers when merging eight ViTs and eight RoBERTa models in Tab. 16.

## E More visualization results

In Section 3, we showed some visualization results using t-SNE [69] and Grad-CAM [61]. Here we provide more visualization results of both existing merging methods and EMR-MERGING. t-SNE and Grad-CAM visualization results are shown in Fig. 8 and Fig. 9, respectively.

# F Configuration of Fig. 4 and Fig. 7

In Fig. 4 and Fig. 7, we hope to compare the sign conflicts, L2 distance, and cosine similarity of the merged model weights and individual model weights. To calculate the sign conflicts, we element-wisely compare the merged model weights to each individual model weights and record the ratio of the elements whose signs conflict. We report the average value of the sign conflicts between the merged model and each individual model. To calculate the L2 distance or cosine similarity, we first flatten the merged model weights and each individual model weights as 1-dimension vectors. Then we calculate the L2 distance or cosine similarity between the merged model and each individual model and report the average value.

# G Limitations and future works

Despite the convincing results, the proposed method suffers from several limitations. On the one hand, compared to existing methods, EMR-MERGING requires a little additional memory to store the light-weight task-specific modulators. On the other hand, as a common limitation of task vector-based methods, EMR-MERGING cannot be generalized to models trained from-scratch because the task vector is based on the pretrain-finetune paradigm.

Further improving the performance of the merged model and generalizing model merging to models trained from-scratch or even models with different structures are significant directions for future work. Additionally, combining model merging with low bit-width quantization has broad application prospects and is also a potential future work.

