# OpenReview forum: "EMR-Merging: Tuning-Free High-Performance Model Merging"
_NeurIPS.cc/2024/Conference — NeurIPS 2024 spotlight_

### Official Review · Reviewer_b5vt · 2024-07-07

**Soundness:** 3
**Presentation:** 2
**Contribution:** 3
**Rating:** 5
**Confidence:** 4

**Summary:**

The paper presents a method called EMR-MERGING (ELECT, MASK & RESCALE-MERGING) for merging models finetuned on different tasks into a single model with multi-task capabilities without the need for additional tuning or training. This method addresses the limitations of existing model merging techniques, which often suffer from performance degradation or require additional data and tuning.

**Strengths:**

- **Theoretical and Empirical Analysis:** The paper provides a solid theoretical foundation for the effectiveness of the proposed method, including detailed proofs and empirical analyses. This adds robustness to the claims made by the authors.
- **Tuning-Free:** One of the major advantages of this method is that it does not require any additional data, tuning, or training, making it highly practical for real-world applications where data availability is limited or privacy concerns restrict access.

**Weaknesses:**

- **Additional Computational Overhead:** The proposed method introduces extra computational overhead during inference due to the use of masks and rescalers. This aspect should be discussed in the paper, particularly regarding its impact on the overall efficiency and performance in real-world applications.
- **Storage Considerations for PEFT Models:** Pre-trained models finetuned on downstream tasks using Parameter-Efficient Finetuning (PEFT) methods typically require only the storage of the pre-trained model weights and additional adapter weights. These adapter weights consume significantly less storage space compared to the masks utilized in EMR-MERGING. The authors should address this comparison and discuss the storage implications in detail.
- **Impact of Model Quantization:** With the trend towards quantization of models, reducing their precision from 32-bit to 8-bit, 2-bit, or even 1-bit, the storage overhead of masks becomes increasingly significant. Quantization is an essential step in the development of large models, and as such, the relative storage cost of the masks in EMR-MERGING will be more pronounced in the future. The authors should consider discussing the impact of model quantization on their method and the potential challenges it poses.
- **Limited Dataset Size in Experiments:** The datasets used in the experiments are relatively small, which undermines the persuasive power of the results. For example, in the case of visual tasks, large models like CLIP's ViT-L exhibit strong generalization capabilities and can achieve high zero-shot classification performance on these small datasets. As a result, the performance differences among various model merging methods might not be significant. The authors should consider using larger, more diverse datasets to better demonstrate the effectiveness and robustness of their approach.

**Questions:**

NA

---

> ### Author Rebuttal · Authors · 2024-08-07
>
> Thank you for your hard work and constructive comments.
>
> ---
> ## Weakness 1: Additional Computational Overhead.
>
> **Ans**: The unified task vector, masks and rescalers are computed during the merging process, which is before the inference process. During the inference process, before we evaluate the performance on a single task, we simply apply the specific mask and rescaler to the unified task vector. This only causes little additional computational costs.
>
> ---
> ## Weakness 2: Storage Considerations for PEFT Models
>
> **Ans**: When merging PEFT models, the task vectors are only generated by the adapters. EMR-Merging only needs to store the masks applied to these adapters because the other weights in the model are fixed. Therefore, compared to storing $N$ adapters, using EMR-Merging to store a unified one and $N$ masks for the adapters can be a strong alternative.
>
> ---
> ## Weakness 3: Impact of Model Quantization
>
> **Ans**:
>
> + The focuses of model merging and Quantization overlap but are not the same.
>
> Model merging focuses on reducing the deployment and storage costs by giving a model multi-task capabilities without training while quantization focuses on reducing computational and storage costs of a single model on a specific task by reducing parameter precision.
>
> + Barriers to combining the model merging and low-bitwidth quantization.
>
> Though combining model merging and low-bitwidth quantization can further reduce the storage costs and computational costs, all the existing model merging methods cannot be directly applied to low-bitwidth quantized models including INT8, INT4, etc. This is due to that model merging needs to subtract or average the weights from different models but after quantization, the low-bitwidth weights cannot be directly operated because these weights may have gone through different quantization functions due to different models' specific weight distribution. Combining model merging with low-bitwidth quantization has broad application prospects and is a potential future work.
>
> + Merging FP-16 models.
>
> In order to explore the applicability of EMR-Merging to FP16 models, we conduct experiments on RoBERTa models loaded in FP16 and the results are shown in Table R12 of the Rebuttal PDF. EMR-Merging still shows promising performance, demonstrating its potential to be applied to quantized models.
>
> + Quantizing the unified task vector.
>
> Although the quantized model cannot be directly merged, we find that the unified task vector can be simply quantized. We layer-wisely convert the elements in the unified task vector into low-bitwidth ones linearly. We use this method to merge RoBERTa models and the results are shown in Table R13. We find that EMR-Merging 's performance is not affected when the unified task vector is quantized to 4-bit and is still promising when the unified task vector is quantized to 2-bit. This can further reduce the storage cost of EMR-Merging.
>
> + The increasingly significant storage overhead of masks.
>
> A potential limitation of EMR-Merging is that, with the trend towards quantization of models, the storage of the masks can be increasingly significant. We will add this to the Limiataion section.
> However, compared to storing multiple individual non-1-bit quantized models, the storage costs by the masks (1-bit) and rescalers (a single value) are still fewer.
> When applied to merge multiple models, EMR-Merging may still tend to be a competitive alternative.
> We will add this discussion to the Future Work section.
>
> ---
> ## Weakness 4: Limited Dataset Size in Experiments
>
> **Ans**:
> + The good performance of EMR-Merging is not merely due to the strong zero-shot capabilities of CLIP models because other merging methods normally suffer from significant performance degradation compared to individual models while EMR-Merging can achieve the performance close to individual models.
> + We have conducted some experiments on large-scale datasets including COCO, ImageNet, and VQA v2 when merging multiple BEiT3 models and EMR-Merging demonstrates its good performance on all these large-scale datasets.
> + Your opinion that using a larger and more diverse dataset may better demonstrate the effectiveness of EMR-Merging is reasonable. Based on merging eight vision tasks using CLIP ViT-B/32, we add a task, i.e., ImageNet-1K. We downloaded a CLIP ViT-B/32 model finetuned on the task in timm and merged the nine models. The results are in General Response, Additional Result #2. EMR-Merging shows a much more significant improvement compared to existing merging methods (up to 20%). We will release this benchmark as a new model merging benchmark.

---

> ### Comment · Reviewer_b5vt · 2024-08-09
>
> The author seems to have misunderstood my points regarding Weaknesses 2 and 3. I would like to clarify them as follows:
> - Weakness 2: A pre-trained model with N adapters can also perform multiple tasks. In this case, I don't quite understand why model fusion is necessary. What advantages does the proposed EMR-MERGING method have over the PEFT approach? PEFT allows for fine-tuning downstream tasks while reducing storage space requirements and achieving satisfactory results. Furthermore, PEFT consumes significantly less storage space compared to storing a mask.
> - Weakness 3: In cases of highly quantized models (e.g., 8-bit), EMR-MERGING would occupy 1/8 of the original model’s size, which significantly limits the effectiveness of the method. In contrast, PEFT does not encounter this issue.

---

> ### Author Response · Authors · 2024-08-09
> **Rebuttal by Authors**
>
> Thank you so much for your review and feedback. We further address your comments regarding Weakness 2 and Weakness 3 as follows:
>
> Answer to Weakness 2:
>
> 1) **Model finetuning and model merging are fundamentally different tasks with different targets**. For model fine-tuning, it consists of full-parameter fine-tuning that pursues fine-tuning performance and parameter-efficient fine-tuning (PEFT) that pursues fine-tuning efficiency, both of which have the same target of customizing models for specific domains, **necessitating labeled datasets and sufficient computational resources for supervised training**. For model merging, under the background that there exists an exponentialy increasing number of pre-trained or finetuned model weights, it targets to take advantage of these existing weights and obtain a single model with multi-task abilities, **without the need for labeled data or supervised training**.
>
> 2) **Model merging can be combined with kinds of model fine-tuning methods**. More specifically, the proposed **EMR-Merging can be applied to not only fully finetuned models but also PEFT models**, as shown in Tab. 6 of our paper. We only need the fine-tuned adapters on each task, and EMR-Merging can merge these adapters into a single adapter with a few masks and rescalers, whose overhead is less than storing all the adapters while achieving performance close to that of multiple individual adapters.
>
> 3) **Model finetuning and model merging are both significant techniques**. In this paper, our EMR-Merging focuses on realizing tuning-free and high-performance model merging instead of reducing storage space requirements as much as possible like PEFT. Besides, **model merging and PEFT can be easily combined to achieve further parameter reduction**, which is a potential area for future work. For example, there have been some studies focusing on merging PEFT models [1,2].
>
> Answer to Weakness 3:
>
> 1) When **compared to multiple individual 8-bit models**, EMR-Merging that consists of one 8-bit model and several masks and scalars can still significantly reduce the parameter numbers.
>
> 2) When **compared to other merging methods**, EMR-Merging shows significantly better performance requring no tuning, demonstrating its greater applicability.
>
> 3) **A potential solution to your concern is using PEFT techniques to train the quantilized models and merge the PEFT modules using EMR-Merging**. This can minimize the number of parameters on multiple tasks, which will be included in our future work.
>
> ----
> **References**
>
> [1] Parameter efficient multi-task model fusion with partial linearization, ICLR 2024.
>
> [2] Composing Parameter-Efficient Modules with Arithmetic Operations, NeurIPS 2023.

---

> > ### Comment · Reviewer_b5vt · 2024-08-14
> >
> > Thank you for your responses,  I appreciate the effort and detail you have provided in addressing the concerns raised.
> >
> > I have no additional questions at this time and will determine my final score after conferring with the Area Chair and the other reviewers.

---

> > > ### Author Response · Authors · 2024-08-14
> > > **Thanks for your comments**
> > >
> > > We appreciate your comments, which can help improve our paper's quality greatly and inspire our future work.
> > >
> > > The discussion about the impact of model quantization and PEFT will be included in the revision.
> > >
> > > Thank you for your efforts. You are welcome to discuss with us if you have any other questions.

---

### Official Review · Reviewer_Qs8J · 2024-07-07

**Soundness:** 3
**Presentation:** 3
**Contribution:** 3
**Rating:** 5
**Confidence:** 5

**Summary:**

The authors identify two issues in current model merging methods: significant performance degradation from the multi-task counterpart and requirement of additional data and training. To tackle those issues, they propose EMR merging, which first creates a unified task vector, then selects masks based on each task-specific model’s correspondence with the unified task vector, finally rescales it. EMR merging shows strong empirical performance among several benchmarks.

**Strengths:**

1. The paper is well written and the algorithm is coupled with adequate amount of analysis.
2. The proposed algorithm is computationally lightweight, yet obtains significant performance improvement.

**Weaknesses:**

1. The novelty is limited. The three steps are essentially a combination of TIES and DARE, where only the masking step is slightly different from TIES. Due to this high similarity, it is unclear why the proposed method performs so much better than TIES and DARE. It would be better to provide a more disentangled analysis for each of the three steps through ablation studies, so that the readers can have a better understanding where exactly the performance improvement comes from.
2. Several related works are missing [1,2] (both on ArXiv in Feb 2024). It is important to do a thorough comparison, especially when the paper basically claims SoTA results.
3. Not all methods are compared on all benchmarks. For instance, AdaMerging is not compared with on NLP benchmarks. Although the original paper of AdaMerging does not provide results in NLP, I do not see any algorithmic obstacle that prevents it from being implemented in the NLP setting.

[1] Representation Surgery for Multi-Task Model Merging. Yang et al. ICML 2024.

[2] Merging Multi-Task Models via Weight-Ensembling Mixture of Experts. Tang et al. ICML 2024.

**Questions:**

1. What is the sparsity of the masks? I know that it is kind of task/model dependent, but it would be nice to show those results.
2. Do EMR still need the hyperparameter tuning on the scaling factors? Or the one from rescalder is final? Because in DARE, they tune the scaling factors even after they compute the rescaling factors.

**Limitations:**

The authors adequately addressed the limitations.

---

> ### Author Rebuttal · Authors · 2024-08-07
>
> Thank you for your hard work and helpful comments.
>
> ---
> ## Weakness 1: limited novelty. The three steps are a combination of TIES and DARE.
>
> **Ans**: The proposed method is totally different from DARE+TIES-Merging.
>
> + The motivation for EMR-Merging is different from existing methods.
>
> We first decouple model merging into unified parts and task-specific parts. Existing model merging methods can be formulated as: $W_M = \mathcal{M}\left(\left[ W_1..W_N \right]\right)$, while our EMR-Merging can be formulated as: $W_{uni}, \left[ E_1..E_N \right] = \mathcal{M'}\left(\left[ W_1..W_N \right]\right)$.
> We elect the unified task vector to reserve the most significant elements and use the masks and rescalers to align the signs and the magnitudes of the merged model, thus bringing about significant improvements. As Reviewer Lqs2 noted, the idea of electing a unified model and identifying different modulators for different tasks is interesting because directly merging models can lead to function conflicts. Differently, TIES and DARE are both proposed to reduce interference but a single model may not optimally simulate the task-specific model weights on all the tasks.
>
> + The process of EMR-Merging is different from TIES and DARE.
>
> TIES and DARE drop partial elements based on magnitude or randomly. The goal of dropping is to reduce interference while our EMR-Merging can avoid sign conflicts by pre-computed masks. DARE's rescaling is to compensate for the dropping process while ours is to align each task-specific model. Additionally, the resclaers for DARE are normally larger than 1 while the rescalers for EMR-Merging are normally smaller than 1 as shown in Table R8 of the Rebuttal PDF. This reflects the difference in the functions of the rescalers of EMR-Merging and DARE.
>
> + EMR-Merging shows tuning-free and plug-and-play properties.
>
> TIES+DARE introduces multiple hyper-parameters while EMR-Merging requires no hyper-parameter tuning, making it highly practical for real-world applications (Reviewer b5vt).
> EMR-Merging also shows plug-and-play capabilities and the performance of EMR-Merging has been verified under vision, NLP (including PEFT and FFT), and multi-modal settings.
>
> + Experimental comparisons.
>
> We additionally compare the performance of EMR-Merging and TIES+DARE in Table R11 of the Rebuttal PDF. It can be seen that the performance TIES+DARE is sensitive to hyper-parameters and our tuning-free EMR-Merging achieves better performance.
>
> + The reason for the improvement.
>
> As to the reason for the performance improvement, we believe that it is because EMR-Merging uses masks to avoid sign conflicts and rescalers to lower the L2 distance between the merged model and task-specific ones. Please check Appendix B for more information. Additionally, we have conducted ablation studies on all three steps of EMR-Merging in Tab. 8 and 9 of our paper. The results show that: 1) exiting merging methods can obtain performance improvement through Mask and Rescale. 2) Both masking and rescaling can boost the performance of the elected vector.
>
> ---
> ## Weakness 2: comparison to the WEMoE [R3] and Representation Surgery [R4].
>
> **Ans**: It should be noted that we did not compare EMR-Merging to WEMoE [R3] and Representation Surgery [R4] in our paper because they had not been officially published by ICML when we submitted the paper.
>
> We compare the parameter numbers and prerequisites of WEMoE, Representation Surgery, and EMR-Merging in Table R1 of the Rebuttal PDF. The proposed EMR-Merging needs no data, training, or tuning while WEMoE and Surgery require unlabeled test data to train partial parameters in their frameworks. Meanwhile, EMR-Merging achieves promising performance. The results of merging ViTs on the eight vision tasks are shown in Table R2 of the Rebuttal PDF. EMR-Merging achieves the SOTA performance on the benchmark of merging ViT-L/14 models. Moreover, EMR-Merging can be easily extended to other settings including NLP and multi-modal.
>
> ---
> ## Weakness 3: not all methods are compared on all benchmarks.
>
> **Ans**: AdaMerging needs to train the merging coefficients. Shifting it to NLP settings requires re-building the framework and re-training. Since AdaMerging is only implemented in vision settings officially and its implementation in other settings is not yet realized by some newly-released toolkits and benchmarks including MergeKit [R1] and FusionBench [R2], we only compare the performance of AdaMerging under vision settings. We argue that the rebuttal time is very limited and re-building and re-training all the methods including AdaMerging under all the benchmarks is beyond the scope of this paper.
>
> However, we are glad to release the benchmarks, checkpoints, and the code of EMR-Merging under all the experimental settings. We have released all the checkpoints and datasets used in our experiments in the anonymous github repository of the manuscript. We have released the code under vision settings and we will release the code for all the other experimental settings soon.
>
> ---
> ## Question 1: the sparsity of the masks.
>
> **Ans**: We show some results of the sparsity of the masks under different settings in Table R5, R6, and R7 of the Rebuttal PDF.
>
> ---
> ## Question 2: does EMR-Merging still need the hyperparameter tuning on the scaling factors?
>
> **Ans**: We promise that EMR-Merging requires **no tuning on any hyper-parameter under any experimental setting**. Actually, EMR-Merging has no hyper-parameters. The unified task vector and the task-specific modulators are all calculated using model weights. We have released the anonymous code under vision settings and we welcome you to reproduce our experiments.
>
> ---
> ## References
>
> [R1] Arcee's MergeKit: A Toolkit for Merging Large Language Models.
>
> [R2] FusionBench: A Comprehensive Benchmark of Deep Model Fusion.
>
> [R3] Merging Multi-Task Models via Weight-Ensembling Mixture of Experts.
>
> [R4] Representation Surgery for Multi-Task Model Merging.

---

### Official Review · Reviewer_S3RQ · 2024-07-09

**Soundness:** 3
**Presentation:** 3
**Contribution:** 2
**Rating:** 5
**Confidence:** 4

**Summary:**

Model merging directly fuses multiple independently trained models at the weight level to obtain a single comprehensive model, which is a current research hotspot. This paper proposes a new model merging method EMR-MERGING to reduce the performance gap between the merged model and the independent models.

**Strengths:**

- The proposed EMR-MERGING method does not require additional training, data, and tuning, which enhances the applicable scenarios of the method.
- This paper conducts extensive experiments, including visual models, NLP models, and multi-modal models. In particular, the visual model is extended to 30 tasks.
- This paper has a clear structure and the proposed methods are easy to implement.

**Weaknesses:**

- **Loss of parallel ability**: Existing model merging methods obtain a unified model for all tasks, so samples from all tasks can be included in one inference (i.e., a batch of data). However, EMR-MERGING needs to configure an independent model for each task during inference, thus losing the ability of parallel inference. In real scenarios, a batch of data may come from different tasks rather than a single task.
- **Important baseline missing**: It is unclear what the benefits and advantages of EMR-MERGING are compared to "sparse independent models". Ties-Merging shows that removing 90% of the neurons in each independent model has almost no change in performance. So why not just use the sparse version of the task vector instead of using EMR-MERGING?
- **Symbol or spelling error**:
  - The symbols are not uniform, and the uniform vector is sometimes $T_F$ and sometimes $T_{uni}$.
  - Table 11: “huper-parameter settings” -> “hyper-parameter settings”
  - Sec 4.2.2: “using 1eleven datasets” -> “using eleven datasets”

- Finally, there is a lack of datasets and checkpoints for CV (30 tasks), NLP, and multimodal in the anonymous link. If the authors can release them, it will help the community research.

**Questions:**

- Is it possible to select only the maximum value in the Electing operation? Is it also possible to select the average value?
- In Figure 4(c), regarding Cos similarity, TV > Ties-Merging > AdaMerging, but in terms of accuracy, TV < Ties-Merging < AdaMerging. Figure 4(b) is similar. These phenomena seem to be contradictory. How does the author explain this?

**Limitations:**

The authors discuss some limitations of the method: for example, it consumes additional memory compared to other model merging methods, and all rely on the same pre-trained model. However, as in Weaknesses, this method may lose the ability to parallelize inference on multiple tasks, and it is unclear how much benefit it has compared to sparse independent models.

---

> ### Author Rebuttal · Authors · 2024-08-07
>
> Thank you for your hard work and kind comments.
>
> ---
> ## Weakness 1: parallel ability.
> **Ans**: 1) Most multi-task model merging methods cannot handle the situation where multi-task samples are included in one inference because only one classification head can be applied during one inference. All the existing merging methods do not merge classification heads [R1-R3] because the classification heads may not share the same architecture. The usual practice of merging methods is applying the proper classification head manually to the merged model before testing on a specific task.  2) We follow this setting and additionally apply a mask and a rescaler to the unified model before testing on a specific task. Once the problem of manual classification head selection is handled, the modulators can also be selected automatically, thus realizing parallel inference. 3) A potential solution is using a gating network to sample-wisely determine which task an input comes from before each inference [R4].
>
> ---
> ## Weakness 2: comparison to multiple Sparse Models
>
> **Ans**: Here we show a detailed comparison.
>
> + The performance of EMR-Merging is more promising.
>
> We compare the performance of Sparse Models and EMR-Merging on merging (2-8) ViT-B/32 models and 30 ViT-B/16 models in Table R3 of the Rebuttal PDF. The $K$ for the Sparse Models refers to keeping the top $K\%$ elements following TIES-Merging [R2]. It can be seen that EMR-Merging performs better than Sparse Models when merging vision models. Additionally, we compare the performance of Sparse Models and EMR-Merging on merging BEiT3 models in Table R4 of the Rebuttal PDF. The performance of Sparse Models severely decreases on some tasks while EMR-Merging shows promising performance on all tasks.
>
> + EMR-Merging leads to smaller storage costs when merging multiple models.
>
> The comparison of parameter numbers is shown in Figure R1 of the Rebuttal PDF.
> EMR-Merging shows both fewer parameter numbers and better performance when compared to Sparse Models (K=10).
> Though Sparse Models (K=5) can realize fewer parameters, their performance is severely reduced.
> It should be noted that the number of parameters in EMR-Merging increases more slowly with the number of tasks due to the lightweight task-specific modules.
> Therefore, when the number of tasks further increases, the parameter numbers of EMR-Merging tend to be fewer than Sparse Models (K=5).
>
> + The elected unified task vector of EMR-Merging can also be sparse.
>
> During EMR-Merging, after we elect $\tau_{uni}$ and before calculating the rescalers, we can make $\tau_{uni}$ sparse by keeping the top $10\%$ ones. This can further reduce the storage cost of EMR-Merging. We identify this method by EMR-Merging (Sparse).
> The performance and parameter numbers of EMR-Merging (Sparse) can be found in Table R2 and Figure R1 of the Rebuttal PDF.
> It can be seen that EMR-Merging (Sparse) sharply reduces the parameter numbers while maintaining high performance, which is still better than Sparse Models while showing the fewest parameter numbers.
>
> ---
> ## Weakness 3: typos.
>
> **Ans**: In the method section, the $\tau_{uni}$ was mistakenly written as $\tau_{F}$. Sorry for that. However, in Theoretical Analysis, we use $\tau_{F}$ to refer to the collection of the merged task vector by existing merging methods and the elected task vector $\tau_{uni}$. This is to demonstrate that regardless of which method is applied for merging (or electing), masking and rescaling can lower the distance between the merged model and individual models, thus explaining why other methods can be combined with Mask & Rescale and get improved in Tab. 8.
>
> To avoid misunderstanding, we will uniformly use $\tau_{uni}$ in the revision. We will also fix the other typos and thanks for pointing them out.
>
> ---
> ## Weakness 4: datasets and checkpoints.
>
> **Ans**: We have released the checkpoints and datasets in the anonymous link of the manuscript. Hopefully, the released checkpoints and newly established benchmarks can help community research.
>
> ---
> ## Question 1:  Is the maximum or the average strategy possible?
>
> **Ans**: Additional ablation studies on the electing process are shown in Table R9 and R10 of the Rebuttal PDF. Applying the maximum strategy may result in better performance on partial tasks including vision tasks and partial multi-modal tasks and worse performance on the rest tasks. In contrast, the average strategy results in worse performance on most tasks.
>
> Note that the main contribution of EMR-Merging is the idea of decoupling model merging into unified parts and task-specific parts. The unified vector can be obtained by not only electing but also existing merging methods, which is shown in Tab. 8.
> EMR-Merging shows tuning-free, data-free, and plug-and-play properties. The effectiveness of EMR-Merging is verified through theoretical and empirical analysis.
>
> ---
> ## Question 2:  the contradictory phenomena.
>
> **Ans**: 1) The proposed EMR-Merging first considered using masks and rescalers to align the signs and magnitudes. Therefore, the sign conflicts can be avoided and the L2 distances can be maximally reduced, as we illustrated in Theoretical Analysis. This explains why EMR-Merging shows the lowest sign conflicts, L2 distance, and the highest cos-sim in Fig. 4. 2) In Fig. 7, we show the sign conflicts, L2 distance, and cos-sim of each step of EMR-Merging. Combined with Tab. 9, it can be seen that during the 3 steps of EMR-Merging, we reduce sign conflicts and L2 distance and increase cos-sim, and the performance is correspondingly increased. 3) Existing methods may not focus on reducing sign conflicts and L2 distances, thus their performance may not correspond with sign conflicts and L2 distances.
>
> ---
> ## References
> [R1] Editing models with task arithmetic
>
> [R2] Ties-merging: Resolving interference when merging models
>
> [R3] Adamerging: Adaptive model merging for multi-task learning
>
> [R4] Merging Vision Transformers from Different Tasks and Domains

---

### Official Review · Reviewer_Lqs2 · 2024-07-11

**Soundness:** 3
**Presentation:** 3
**Contribution:** 3
**Rating:** 6
**Confidence:** 4

**Summary:**

This paper aims to improve the performance of model merging in the multi-task learning domain. By electing a unified model, identifying a mask and a rescaling factor for each task, the proposed method EMR-Merging is able to significantly improve the merging performance over the previous state-of-the-art, such as Adamerging, Ties-Merging, Task Arithmetic, etc. Besides the CV model merging, EMR-Merging also effectively merges the large language models, approaching the individual fine-tuned performance.

**Strengths:**

1. I like the idea that we need to elect a unified model first, and then identify different masks for different tasks. Directly merging models can lead to function conflicts.

2. The experiments are extensive, including VIT model, language models, and multi-modal models, with considerable good performance.

3. The ablation study is also provided.

**Weaknesses:**

1. I am curious why the proposed way works for the unified model. One specific case is that, if the the average of the task vector is close to zero, but not exactly equal to zero, e.g., 0.0001, which will give us a positive sign. If the magnitude of this weight is large, a positive sign multiplied by this large magnitude weight can lead to a large error. Can the authors elaborate more the intuition behind the unified model, mask, and rescaler?

2. Can you explain the process of obtaining the mask? I could not find the definition of $\tau_F$ after reading the paper in detail.

3. What is the configuration of Figure 4? More details are needed here.

4. How to interpret the t-SNE visualization?

5. The distance between figures is over-reduced. It can be confusing and misleading. For instance, figure 5 and figure 4.

6. How expensive to extend EMR-Merging to LLMs, such as LLaMa3?

7. How does the proposed method perform compared to SOTA https://arxiv.org/pdf/2402.00433?

**Questions:**

Please see the above weaknesses.

**Limitations:**

Provided.

---

> ### Author Rebuttal · Authors · 2024-08-06
>
> Thank you for your hard work and admitting the value of our contribution.
>
> ---
> ## Weakness 1: why the proposed way works, what if the average of the task vector is close to zero, and the intuition of the method.
>
> **Ans**: When we elect the sign vector, we first element-wisely add the task vectors and obtain the elected sign, i.e., $\gamma_{uni} = sgn(\sum_{t=1}^{N} \tau_t)$ when merging $N$ task vectors. Then we element-wisely choose the maximum absolute value of each parameter with the sign consistent with $\gamma_{uni}$ and obtain the elected unified task vector $\tau_{uni}$ . It contains significant elements from all the task vectors.
>
> Next, we calculate the masks by: $M_i= \left( \tau_i \odot \tau_{uni} > 0 \right)$ for task $i$.
>
> Finally, we calculate rescalers by $\lambda_i = \frac{sum(abs(\tau_{i}))}{sum(abs(M_i \odot \tau_{uni}))}$. We apply absolute values in both the numerator and denominator, avoiding the situation that the average of a task vector is close to zero. Additionally, the $\tau_{uni}$ contains the maximum values of the elected direction. This will also prevent the denominator from approaching zero. We observe that the rescalers are normally not larger than 1. We provide the values of the rescalers under various settings in Table R8 of the Rebuttal PDF.
>
> When designing EMR-Merging, we hope to reserve the shared and significant features from all the task vectors in the unified task vector $\tau_{uni}$. We reserve the shared features among task vectors by electing the sign vector and reserve the significant features by electing the elements of the largest absolute values with the sign consistent with the sign vector. Ties-Merging shows that sign conflicts cause significant performance degradation. DropOut [R3] and DARE [R1] shows that simply dropping without magnitude alignment can also effect the performance. Inspired by this, we use masks to avoid sign conflicts between the unified task vector and task-specific ones and use rescalers to align the magnitude.
>
> ---
> ## Weakness 2: the process of obtaining the masks; the definition of $\tau_{F}$.
>
> **Ans**:  The mask for task $i$ is calculated by: $M_i= \left( \tau_i \odot \tau_{uni} > 0 \right)$. In the method section, the $\tau_{uni}$ was mistakenly written as $\tau_{F}$ when introducing the process of obtaining the masks. Sorry for the mistake.  However, in the theoretical analysis, we use $\tau_{F}$ to refer to collection of the merged task vector by existing methods (e.g., Ties-Merging) or the elected task vector $\tau_{uni}$. The definition of $\tau_{F}$ is to demonstrate the effectiveness of the masks and rescalers regardless of the electing process. This can explain why other methods can be combined with Mask & Rescale and obtain performance improvement in Table 8.
>
> We find that this may lead to misunderstanding and we will uniformly replace $\tau_{F}$ by $\tau_{uni}$ in the revision.
>
> ---
> ## Weakness 3: the config of Fig. 4.
>
> **Ans**: In Fig. 4, we hope to compare the sign conflicts, L2 distance, and cosine similarity of the merged model weights and individual model weights.
> To calculate the sign conflicts, we element-wisely compare the merged model weights to each individual model weights and record the ratio of the elements whose signs conflit. We report the average value of the sign conflits between the merged model and each individual model.
> To calculate the L2 distance (cosine similarity), we first flatten the merged model weights and each individual model weights as 1-dimension vectors. Then we calculate the L2 distance (cosine similarity) between the merged model and each individual model and report the average value.
>
> We will add the configuration in the revision.
>
> ---
> ## Weakness 4: interpret the t-SNE.
>
> **Ans**: t-SNE is introduced to visualize high-dimensional data. In our paper, we visualize the outputs of the last transformer block (before the classification head). The results can represent the feature extracting capabilities by the separation among the classes. In Fig. 5, EMR-Merging shows the clearest separation among all the merging methods and is close to individual models. This is correspondent with the experimental results.
>
> ---
> ## Weakness 5: the small distance between figures.
>
> **Ans**: Thanks for your advice and we will increase the distance between figures in the revision.
>
> ---
> ## Weakness 6: how expensive to extend to LLMs?
>
> **Ans**: The proposed EMR-Merging can easily be adapted to various settings including merging LLMs.
>
> + In our paper, we show the experimental results when applying EMR-Merging to T0-3B models finetuned by $(IA)^3$, demonstrating the applicability of EMR-Merging to both LLMs and PEFT models.
>
> + To further demonstrate the high performance and broad applicability of EMR-Merging, we apply EMR-Merging to a new benchmark, which merges GPT-2 models (fully-finetuned) on 7 NLP tasks and it achieves the SOTA performance. The performance improvement is up to more than 10%. The results are in General Response.
>
> ---
> ## Weakness 7: comparison to WEMoE [R2]
>
> **Ans**: We did not compare EMR-Merging to WEMoE [R2] in our paper because WEMoE had not been officially published by ICML when we submitted the paper.
>
> We compare the parameter numbers and prerequisites of WEMoE and EMR-Merging in Table R1 of the Rebuttal PDF. The proposed EMR-Merging needs no data, training, or tuning while WEMoE requires unlabeled test data to train partial parameters from the framework. Meanwhile, EMR-Merging achieves promising performance. The results of merging ViTs on the eight vision tasks are shown in Table R2 of the Rebuttal PDF. EMR-Merging achieves the SOTA performance on the benchmark of merging ViT-L/14 models. Moreover, EMR-Merging can be easily extended to other settings.
>
> ---
> ## References
>
> [R1] Language models are super mario
>
> [R2] Merging Multi-Task Models via Weight-Ensembling Mixture of Experts
>
> [R3] Dropout: a simple way to prevent neural networks from overfitting

---

### Author Rebuttal · Authors · 2024-08-07

# General Response:

We thank all the reviwers for their time and constructive comments. We appreciate the reviewers' praise of the strengths of our paper including:

- the idea that decoupling model merging into unified parts and task-specific parts to avoid conflicts (reviewer Lqs2).
- extensive expreiments and considerable good performance (reviewer Lqs2, S3RQ).
- adequate amount of theoretical and empirical analysis (reviewer b5vt, Qs8J).
- tuning-free and data-free properties (reviewer b5vt, S3RQ)

-----

In the responses, we show additional experimental results including:

1. Merging GPT-2 models on seven NLP tasks.

   We merge GPT-2 models finetuned on seven NLP tasks from GLUE benchmark following the settings from FusionBenchmark. EMR-Merging improves the best performance of compared model merging methods by up to 10%. The results are in General Response, Additional Results #1.

2. Merging ViT-B/32 models on nine vision tasks (ImageNet-1K added).

   We additionally add ImageNet-1K to the original eight tasks. We merge nine CLIP ViT-B/32 models and EMR-Merging shows a much more significant improvement compared to existing merging methods (up to 20%). The results are in General Response, Additional Results #2.

3. Detailed comparison with Sparse Models.

   We compare the parameter number and performance of EMR-Merging and Sparse Models. In addition, we find that the elected unified task vector of our method can be sparsed to further reduce parameter numbers without affecting performance much. The results are in Table R3, R4, and Figure R1 in the Rebuttal PDF.

4. More ablation studies on the Electing process.

   We conduct additional ablation studies on the electing process of EMR-Merging. In addition, we find that the elected unified task vector can be quantilized to 4-bit or even 2-bit to further reduce parameter numbers without affecting performance much. The results are in Table R13 in the Rebuttal PDF.

5. Detailed comparison with WEMoE and Representation Surgery.

   We show detailed comparison of EMR-Merging with WEMoE [R1] and Representation Surgery [R2] in parameter numbers, prerequisites, and performance. EMR-Merging shows the SOTA performance on merging ViT-L/14 models and requires no data or training. The results are in Table R1 and R2 of the Rebuttal PDF.



----

# Some Additional Results

## 1. Merging GPT-2 Models on seven NLP tasks

Here we provide the results of EMR-Merging on a new benchmark [R3], merging GPT-2 models on seven NLP tasks from GLUE. All the tasks are evaluate by accuracy. Our EMR-Merging improves the performance of model merging by up to 10%.

| Merging GPT2 Models    | CoLA     | MNLI     | MRPC     | QNLI     | QQP      | RTE      | SST-2    | Avg      |
| :--------------------- | :------- | :------- | :------- | :------- | :------- | :------- | :------- | -------- |
| Individual             | 76.8     | 82.1     | 80.4     | 88.3     | 89.6     | 65.3     | 91.2     | 82.0     |
| Simple Average         | 55.0     | 55.1     | 51.0     | 57.6     | 76.7     | 44.8     | 52.5     | 56.1     |
| Fisher Merging         | 54.8     | 58.0     | 39.5     | 63.3     | 81.5     | 49.1     | 64.7     | 58.7     |
| RegMean                | 61.7     | 70.4     | 65.4     | 69.7     | 78.8     | 56.0     | 79.7     | 68.8     |
| Task Arithmetic        | 68.7     | 68.6     | 69.6     | 70.5     | 81.8     | 47.3     | 83.6     | 70.0     |
| Ties-Merging           | 68.4     | 71.4     | 68.4     | 69.6     | 82.4     | 47.7     | 81.8     | 70.0     |
| **EMR-Merging (Ours)** | **72.8** | **81.1** | **79.2** | **84.8** | **88.1** | **66.5** | **90.3** | **80.4** |

## 2. Merging ViTs on nine vision tasks

On the basis of merging ViTs on eight vision tasks, we additionally add ImageNet-1K to the these tasks. The finetuned model on ImageNet-1k is released by timm. We merge nine CLIP ViT-B/32 models and EMR-Merging shows a much more significant improvement compared to existing merging methods (up to 20%).

| ViT-B/32 on nine vision tasks | SUN397   | Cars     | RESISC45 | EuroSAT  | SVHN     | GTSRB    | MNIST    | DTD      | IN-1k    | Avg      |
| ----------------------------- | -------- | -------- | -------- | -------- | -------- | -------- | -------- | -------- | -------- | -------- |
| Individual                    | 75.3     | 77.7     | 96.1     | 99.7     | 97.5     | 98.7     | 99.7     | 79.4     | 82.0     | 89.6     |
| Averaging                     | 61.8     | 56.4     | 65.9     | 66.2     | 62.7     | 44.5     | 81.8     | 49.0     | 61.5     | 61.1     |
| Task Arithmetic               | 51.8     | 30.9     | 55.8     | 64.3     | 69.0     | 42.2     | 92.7     | 46.8     | 66.6     | 57.8     |
| Ties-Merging                  | 53.3     | 34.1     | 57.0     | 55.8     | 72.3     | 43.2     | 90.5     | 46.5     | 68.9     | 58.0     |
| **EMR-Merging (Ours)**        | **77.0** | **75.2** | **92.9** | **92.7** | **79.7** | **90.2** | **97.6** | **76.2** | **79.8** | **84.6** |

The impressive performance of our EMR-Merging on various benchmarks demonstrates its good performance and applicability.

----

# References

[R1] Merging Multi-Task Models via Weight-Ensembling Mixture of Experts. ICML 2024.

[R2] Representation Surgery for Multi-Task Model Merging. ICML 2024.

[R3] FusionBench: A Comprehensive Benchmark of Deep Model Fusion. arXiv:2406.03280.

---

### Decision · Program_Chairs · 2024-09-25

**Decision:**

Accept (spotlight)

**Comment:**

The paper introduces a new method for model merging, a currently hot research topic, which considers how to directly fuse multiple independently trained models at the weight level to obtain a single comprehensive model. The proposed approach addresses the significant performance degradation from the multi-task counterpart and requirement of additional data and training in existing methods.

Strengths of the paper and the proposed method include:
- the paper is well-written.
- method does not require additional training, data, and tuning, which enhances the applicable scenarios of the method. This makes it practical for real-world applications where data availability is limited or privacy concerns restrict access.
- experiments are extensive, including VIT model, language models, and multi-modal models. In particular, the visual model is extended to 30 tasks.
- informative ablation studies are provided.
- the methods are computationally lightweight and straightforward to implement, yet they perform well
- The paper provides a solid theoretical foundation for the effectiveness of the proposed method, including detailed proofs and empirical analyses. This adds robustness to the claims made by the authors.

Regarding the weaknesses identified by the reviewers, the rebuttal and subsequent discussion effectively answered most of these, including the additional computational overhead during inference and comparison to parameter-efficient fine tuning methods.

The discussion about the applicability of the method to quantized networks was quite fruitful, and the proposed changes to the document based on this will add to the depth of the presentation of the method.

The rebuttal effectively addresses the novelty of the proposed method, showing that it is quite different than the methods indicated by the reviewer.

The issue of not being able to do multiple tasks in parallel is moot, since competing methods cannot do this either.